# SPAE: Semantic Pyramid AutoEncoder for Multimodal Generation with Frozen LLMs

**Lijun Yu**[‡†*]    **Yong Cheng**[†]    **Zhiruo Wang**[‡]    **Vivek Kumar**[†]    **Wolfgang Macherey**[†]

**Yanping Huang**[†]    **David A. Ross**[†]    **Irfan Essa**[†]    **Yonatan Bisk**[‡]    **Ming-Hsuan Yang**[†]

**Kevin Murphy**[†]    **Alexander G. Hauptmann**[‡]    **Lu Jiang**[†‡]

[†]Google, [‡]Carnegie Mellon University

## Abstract

In this work, we introduce Semantic Pyramid AutoEncoder (SPAE) for enabling frozen LLMs to perform both understanding and generation tasks involving non-linguistic modalities such as images or videos. SPAE converts between raw pixels and interpretable lexical tokens (or words) extracted from the LLM's vocabulary. The resulting tokens capture both the semantic meaning and the fine-grained details needed for visual reconstruction, effectively translating the visual content into a language comprehensible to the LLM, and empowering it to perform a wide array of multimodal tasks. Our approach is validated through in-context learning experiments with frozen PaLM 2 and GPT 3.5 on a diverse set of image understanding and generation tasks. Our method marks the first successful attempt to enable a frozen LLM to generate image content while surpassing state-of-the-art performance in image understanding tasks, under the same setting, by over 25%.

## 1 Introduction

Large language models (LLMs) empowered by Transformers [38] have achieved remarkable progress in addressing a broad spectrum of Natural Language Processing (NLP) tasks [4, 8, 28, 2]. With the continuous increases in model size and training data, LLMs are gradually becoming more versatile and agnostic to specific tasks, unlocking new capabilities in solving complex AI tasks [42], like question answering, code generation, reasoning, mathematics problem-solving, and understanding humor, among various other applications [2, 28].

LLMs capture rich conceptual knowledge about the world in their lexical embeddings. This raises a question: if provided with the appropriate visual representations as input, *are frozen LLMs capable of solving tasks in visual modalities?* Very recently, there have been notable advancements in extending the capabilities of frozen LLMs to tackle image understanding and retrieval tasks [21, 27]. However, generating a different modality using a frozen LLM that has not been explicitly trained on that modality has proven to be challenging and has had little success.

To facilitate LLMs for such cross-modal tasks, we propose to learn a vector quantizer to map an image, or some other non-linguistic ("foreign") modality, to the token space of a frozen LLM. This effectively translates the image into a language that the LLM can comprehend, enabling us to leverage the generative abilities of the LLM to perform image understanding and generation tasks without having to train on image-text pairs. Specifically, our new approach is that, given an image prompt, convert it to a token space with our learned encoder, use the LLM to generate suitable lexical tokens, and convert back to pixel space with our learned decoder.

---

[*]Work partially done during a research internship at Google Research.

37th Conference on Neural Information Processing Systems (NeurIPS 2023).

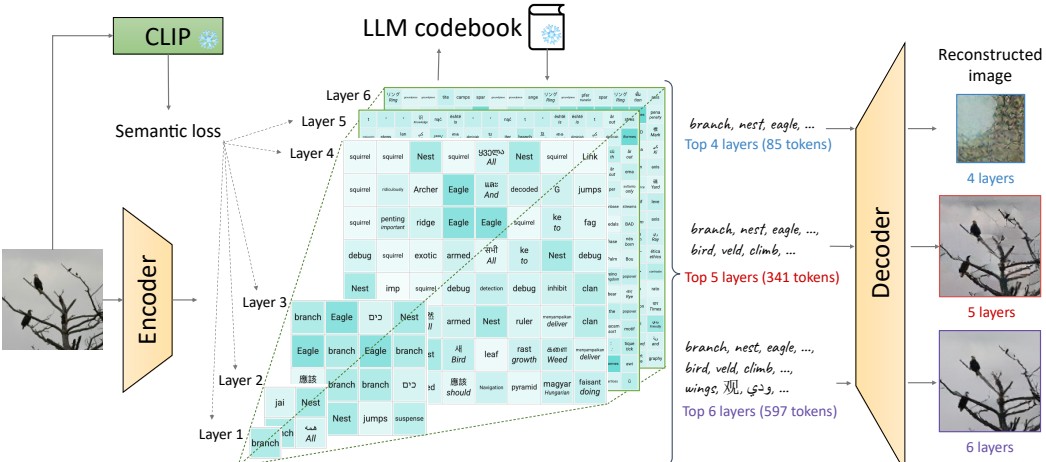

Figure 1. **Framework of the proposed SPAE model.** An image is encoded into a pyramid of lexical tokens capturing semantic concepts and appearance details necessary for reconstruction.

We introduce a novel Semantic Pyramid AutoEncoder (SPAE) that produces a lexical word sequence that (1) carries rich semantics, and (2) retains fine details for signal reconstruction. In contrast to the majority of VQ-VAE approaches [37], our encoder maps to an interpretable discrete latent space, *i.e.*, words. As depicted in Fig. 1, SPAE tokens have a multi-scale representation arranged in a pyramid structure. The upper layers of the pyramid comprise semantic-central concepts, while the lower layers prioritize appearance representations that captures the fine details for image reconstruction. This design enables us to dynamically adjust the token length to accommodate various tasks, such as using fewer tokens for understanding tasks and more tokens for generation tasks.

We verify the plausibility of our approach in an extreme setting of in-context learning [4], without any parameter updates to the LLM. Our SPAE model is trained standalone, without backpropagating through any language model. We evaluate our approach on image understanding tasks including image classification, image captioning, and visual question answering. We showcase a promising direction to image generation with LLMs by utilizing in-context denoising techniques. Our method is LLM-agnostic and has been tested with PaLM 2 [2] and GPT-3.5 [28], suggesting compatibility with arbitrary LLMs.

The main contributions of this work are summarized as follows:

- This is the first successful method, to the best of our knowledge, that uses a frozen language model, trained solely on language tokens, to directly generate image content through in-context learning.
- We introduce a new SPAE tokenizer producing interpretable representations of semantic concepts and fine-grained details in the form of multilingual linguistic tokens with adjustable lengths.
- We evaluate our method on visual understanding and generation tasks, and notably, our approach outperforms the best-published few-shot image classification accuracy [27] by an absolute 25% under the same in-context setting.

## 2 Related Work

**Multimodal generation with LLMs.** Advances have been made to expand the capabilities of LLMs beyond language. For example, Visual ChatGPT [43] uses ChatGPT to generate prompts and executes multimodal tasks through another model, *e.g.*, generating image from text prompts by Stable Diffusion [32]. FROMAGe [21] feeds CLIP [30] embeddings to OPT [49] for image understanding and retrieval. However, it requires backpropagation through the LLM and does not support image synthesis. This work enables a standalone frozen LLM to understand and generate other modalities which are unseen in training.

**Tokenization via vector quantization.** VQ-VAE [37] compresses data into a discrete latent space defined by a codebook via vector quantization. VQGAN [14] enhances the reconstruction quality with adversarial and perceptual objectives. These discrete latent quantities, often referred to as *tokens*, are widely used to learn generative transformer models for image [32, 7], video [45, 15, 39], image-video [46], and audio [3, 9]. Our SPAE model is built upon the VQGAN framework and applicable to different modalities.

**Tokenization into lexical representations.** The codebooks in typical VQGANs are learned jointly with the encoder and decoder stacks, which are not directly interpretable via natural languages. LQAE [27] replaces the learned codebook with frozen word embeddings from BERT [12] to connect with an English vocabulary. However, the LQAE tokens seldom contain semantic concepts in an image, and the reconstruction quality is worse than that with a learned codebook. Our SPAE quantizes an input sample into semantically related tokens in a multilingual vocabulary while preserving the high reconstruction quality of a VQGAN for generative tasks. In addition, SPAE tokens are organized in a multi-layer coarse-to-fine pyramid for flexible usage in different tasks.

**Few-shot learning with LLMs.** In-context learning [4, 8, 2] facilitates LLMs for few-shot learning via the text interface without parameter updates. This approach is commonly employed to assess the performance of LLMs on numerous NLP benchmarks, *e.g.*, classification and question answering [41], mathematical reasoning [24], and code generation [44], which yields competitive results to their fine-tuned counterparts. However, existing few-shot vision-language understanding and generation frameworks [1, 21] still require LLM parameter updates. In contrast, our work inherits the in-context learning ability from frozen LLMs.

## 3 Method

Our goal is to model an image, or some other non-linguistic modality (*e.g.*, video or audio), as a language sequence that LLMs can comprehend. *Semantic Pyramid AutoEncoder* (SPAE) generates a lexical word sequence with dynamically adjustable length that carries rich semantics and retains fine details for signal reconstruction. To work with a frozen LLM via in-context learning, we introduce a progressive in-context denoising method to facilitate image generation. We use the image modality in this section to introduce our SPAE model in 2D, and later showcase the results of a 3D variant with the video modality in our experiments.

### 3.1 Semantic Pyramid AutoEncoder

Our SPAE model extends the VQ-VAE [37] framework, which comprises an encoder, a quantizer, and a decoder. The CNN encoder maps an image $\mathbf{I} \in \mathbb{R}^{H \times W \times 3}$ into continuous embeddings $\mathbf{Z} \in \mathbb{R}^{h \times w \times c}$. Each element $\mathbf{z} \in \mathbf{Z}$ is then passed through the quantizer, which assigns it to the closest entry in a codebook, resulting in the quantized embedding. Let $\hat{\mathbf{Z}}$ represent the quantized embeddings for the entire image. The CNN decoder receives $\hat{\mathbf{Z}}$ as input and generates the reconstructed image $\hat{\mathbf{I}}$. Below we highlight the design differences in SPAE.

As illustrated in Fig. 1, SPAE generates lexical tokens arranged in a pyramid structure, which contains semantic concepts in the upper layers and appearance with progressively refined details in the lower layers. We introduce a semantic loss to encourage the usage of conceptually relevant tokens.

**Frozen language codebook.** To generate lexical tokens, we utilize a pretrained LLM codebook $\mathbb{C} = \{(k, \mathbf{e}(k)) \mid k \in \mathbb{T}\}$ and freeze it during training, where $\mathbb{T}$ is a subset of the LLM vocabulary. Here, $\mathbf{e}(\cdot)$ produces the text embedding for a sub-word $k$ which may be obtained from any layer of the LLM. Since the codebook is aligned with the language vocabulary, we use the terms "token" and "word" interchangeably.

**Token pyramid.** The SPAE quantizer produces $D$ layers of tokens where the tokens at layer $l$ are denoted as $\mathbf{k}_l \in \mathbb{T}^{h_l \times w_l}$. Prior works use Residual Quantization (RQ) to generate multi-layer tokens [22, 47]. In these methods, tokens from all layers have uniform shapes and do not carry specific semantic meanings. In contrast, we propose a pyramid token structure by enforcing the constraint $h_l \leq h_{l+1} \wedge w_l \leq w_{l+1}$. The pyramid structure is purposefully designed to concentrate semantics within the within the upper layers of the pyramid. This design allows for representing semantic concepts with notably fewer tokens, *e.g.*, as few as five tokens for understanding tasks. The high token efficiency stems from the pyramid structure, as a conventional layer without pyramid structures needs a minimum of $hw$ tokens (*e.g.*, 256) to represent the image. Token efficiency is crucial for in-context learning as it enables the accommodation of more examples within the context. A dilation subsampler $\mathbf{P}(l)$ is used, which selects the positions for quantization at layer $l$ as

$$\mathbf{P}(l) = \{(h'i - \left\lceil \frac{h'}{2} \right\rceil + 1, w'j - \left\lceil \frac{w'}{2} \right\rceil + 1) \mid (i,j) \in ([1, h_l] \times [1, w_l]) \cap \mathbb{Z}^2\}, \quad (1)$$

where $h' = \frac{h_D}{h_l}$, and $w' = \frac{w_D}{w_l}$ are the downsample ratios.

For each embedding $\mathbf{z}$ at position $(x, y)$, we obtain its discrete tokens sequentially from layer 1 to $D$. At layer $l$, if $(x, y) \in \mathbf{P}(l)$, the quantizer assigns discrete token $k_l = \arg\min_{k \in \mathbb{T}} \|\mathbf{z}_l - \mathbf{e}(k)\|_2^2$,

where $\mathbf{z}_l$ is the current layer embedding, calculated from

$$\mathbf{z}_l = \mathbf{z} + \sum_{i=1}^{l-1} \mathbf{1}_{(x,y)\in\mathbf{P}(i)}(\mathbf{z} - \mathbf{e}(k_i)). \tag{2}$$

The quantized embedding reconstructed with the first $l$ layers is given by the average of the existing token embeddings as

$$\hat{\mathbf{z}}_{\leq l} = \frac{\sum_{i=1}^{l} \mathbf{1}_{(x,y)\in\mathbf{P}(i)}\mathbf{e}(k_i)}{\sum_{i=1}^{l} \mathbf{1}_{(x,y)\in\mathbf{P}(i)}}. \tag{3}$$

Using the input of $\hat{\mathbf{Z}}_{\leq l}$ from tokens up to layer $l$, the decoder can progressively reconstruct the image with dynamic token lengths, resulting in gradually improved quality with refined appearance details. We term this approach as *Streaming Average Quantization* (SAQ) due to its resemblance to computing the average on streaming data, where $\hat{\mathbf{z}}_{\leq l+1} = \hat{\mathbf{z}}_{\leq l} + \frac{1}{\hat{l}+1}\mathbf{e}(k_{l+1}), \hat{l} = \sum_{i=1}^{l} \mathbf{1}_{(x,y)\in\mathbf{P}(i)}.$

RQ [22, 47] is applicable but yields worse results in this context, as revealed by our ablation studies. This can be attributed to (1) varying scales of embeddings in residual layers, potentially dividing the codebook into multiple parts, and (2) misalignment in the summation of word embeddings, which undermines learning semantically meaningful tokens in later layers.

**Semantic loss.** We encourage the semantic similarity between the image $\mathbf{I}$ and each lexical token $k$ denoted by $s(\mathbf{I}, k)$. During training, we build per-layer candidate token pools as

$$\mathbf{C}_l(\mathbf{I}) = \{k \in \mathbb{T} \mid s(\mathbf{I}, k) \geq \rho_l\}, \tag{4}$$

where $\rho_l$ is a threshold. We set $\rho_l \geq \rho_{l+1}$ to allow deeper layers to have a larger pool of candidate tokens while sacrificing some semantics.

To define the similarity score, this paper employs a pretrained CLIP model [29]. In more details, let $f_{\mathcal{I}}$ and $f_{\mathcal{T}}$ be a pair of image and text CLIP embedding functions. We precompute the text feature for each token $k \in \mathbb{T}$ as

$$f'_{\mathcal{T}}(k) = \frac{1}{|\mathbf{p}|} \sum_{i=1}^{|\mathbf{p}|} f_{\mathcal{T}}(\mathbf{p}_i(k)), \tag{5}$$

where $\mathbf{p}$ is a list of prompt templates, such as "a photo of ...". During training, we extract the image feature $f_{\mathcal{I}}(\mathbf{I})$ and compute the dot-product similarity as $\mathbf{s}'(\mathbf{I}, k) = f_{\mathcal{I}}(\mathbf{I}) \cdot f'_{\mathcal{T}}(k)$. The similarity score is then normalized to account for the varying scales across different images.

$$\mathbf{s}(\mathbf{I}, k) = \frac{\mathbf{s}'(\mathbf{I}, k) - \min_j \mathbf{s}'(\mathbf{I}, j)}{\max_j \mathbf{s}'(\mathbf{I}, j) - \min_j \mathbf{s}'(\mathbf{I}, j)}. \tag{6}$$

We define the semantic loss for the encoder parameters $\theta_e$ as

$$\mathcal{L}_{\text{semantic}}(\theta_e; \mathbf{I}) = \underset{l\in[1,D']}{\mathbb{E}} \underset{\mathbf{z}_l}{\mathbb{E}} \underset{c\in\mathbf{C}_l(\mathbf{I})}{\mathbb{E}} -\log \frac{\exp(-\|(\mathbf{z}_l - \mathbf{e}(c)\|_2^2)}{\sum_{k\in\mathbb{T}} \exp(-\|\mathbf{z}_l - \mathbf{e}(k)\|_2^2)}, \tag{7}$$

where we randomly sample semantically similar target codes $c$ for each layer embedding in the first $D'$ layers.

**Appearance loss.** Using an improved objective from [45], the appearance loss is calculated as:

$$\mathcal{L}_{\text{appearance}}(\theta_e, \theta_d; \mathbf{I}) = \|\mathbf{I} - \hat{\mathbf{I}}\|_2^2 + \beta \sum_{l=1}^{D} \|\mathbf{Z} - \text{sg}(\hat{\mathbf{Z}}_{\leq l})\|_2^2 + \lambda \mathcal{L}_{\text{GAN}} + \eta \mathcal{L}_{\text{Perceptual}} + \phi \mathcal{L}_{\text{LeCAM}}, \tag{8}$$

where $\mathcal{L}_{GAN}$, $\mathcal{L}_{Perceptual}$, and $\mathcal{L}_{LeCAM}$ are the VQGAN [15], perceptual [19], and LeCAM [34] losses. In addition, $\text{sg}(x)$ is the stop-gradient operation. The appearance loss is applied to both the encoder $\theta_e$ and decoder parameters $\theta_d$, excluding the frozen codebook embedding.

To stabilize the training and balance between appearance and semantics, we add a dynamic weight for the semantic guidance loss as $w = \text{sg}\left(\frac{\mathcal{L}_{\text{appearance}}(\mathbf{I})}{\mathcal{L}_{\text{semantic}}(\mathbf{I})}\right)$. The total training loss excluding the GAN discriminator is

$$\mathcal{L}_{\text{SPAE}}(\theta_e, \theta_q) = \underset{\mathbf{I}}{\mathbb{E}} \left[ \mathcal{L}_{\text{appearance}}(\theta_e, \theta_q; \mathbf{I}) + \alpha w \mathcal{L}_{\text{semantic}}(\theta_e; \mathbf{I}) \right]. \tag{9}$$

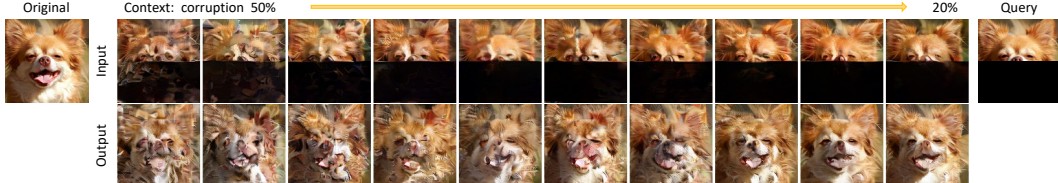

Figure 2. **An example of the conditional image denoising task** for high resolution synthesis. The context comprises images randomly corrupted in the token space.

## 3.2 Progressive In-Context Decoding

While our method is more effective when backpropagating through LLMs by prompt [23] or adapter tuning [17, 18], this work focuses on verifying the plausibility in an extreme setting of in-context learning [4]. We demonstrate that LLMs are capable of performing new tasks in foreign modalities without any parameter updates. Specifically, a set of $K$ examples $\{(\mathbf{u}^i, \mathbf{v}^i)\}_{i=1}^K$ are fed to the LLM to learn a new task and answer a query $\hat{\mathbf{u}}$ with

$$\hat{\mathbf{v}} \sim \mathbb{P}_{\text{LLM}}(\cdot \mid \hat{\mathbf{u}}; \{(\mathbf{u}^i, \mathbf{v}^i)\}_{i=1}^K). \tag{10}$$

Sampling $\hat{\mathbf{v}}$ by a single-pass autoregressive decoding is suboptimal due to the distributional shift in the representation and the presence of exceptionally long sequences, *e.g.*, an image is quantized into over 500 tokens. To this end, we use a progressive decoding method.

We generalize Eq. (10) into a multi-pass process, where the LLM learns to generate one segment of the target sequence at a time. The segment generated from the $t$-th pass is

$$\hat{\mathbf{v}}_t \sim \mathbb{P}_{\text{LLM}}(\cdot \mid [\hat{\mathbf{u}}, \hat{\mathbf{v}}_{<t'}]; \{([\mathbf{u}^i, \mathbf{v}^i_{<t'}], \mathbf{v}^i_t)\}_{i=1}^K), \tag{11}$$

where $[\cdot, \cdot]$ indicates concatenation. $t'$ controls the length of previous segments to condition on, with two common cases: (1) a progressive autoregressive (PAR) process with $t' = t$, where each decoded segment conditions on all previously decoded ones; (2) a progressive non-autoregressive (PNAR) process with $t' = 0$ to sample each segment independently, which greatly reduces the sequence length requirement for the LLM. In practice, we use PAR to generate the first few token layers given task-specific conditions, followed by PNAR to generate the remaining token layers conditioned on the previous layers in an unconditional latent refinement process.

The learning capacity of an in-context setup is far from sufficient for a modality that has not been seen during training. So far, there have been no successful attempts in the literature demonstrating that a frozen LLM can generate image content. For low-resolution images, LLMs can produce images directly using in-context learning, as will be demonstrated with $32 \times 32$ MNIST images [11]. For higher resolutions, the context length restricts the number of examples. For instance, a context window of 8k tokens can only hold less than a dozen $128 \times 128$ images. Therefore, we operate in a denoising subspace to synthesis beyond $32 \times 32$ resolution. Fig. 2 illustrates one example, with detailed definitions in the Appendix.

## 4 Experimental Results

### 4.1 Experimental Settings

To verify the compatibility with different LLMs, we train two variants of SPAE, namely SPAE$_{\text{PaLM}}$ and SPAE$_{\text{GPT}}$. The SPAE$_{\text{PaLM}}$ codebook is taken from the input embedding layer of a PaLM 2-S checkpoint with a 65k vocabulary of the most frequent sentence pieces. The PaLM 2-L API [2] is used for in-context learning with SPAE$_{\text{PaLM}}$. SPAE$_{\text{GPT}}$ uses a byte-pair encoding vocabulary with 99k UTF-8 tokens (https://github.com/openai/tiktoken), where we obtain the contextual token embeddings from OpenAI `text-embedding-ada-002` (https://platform.openai.com/docs/models/embeddings). For a fair comparison with prior works [27], we use SPAE$_{\text{GPT}}$ with the GPT 3.5 `text-davinci-003` API (https://platform.openai.com/docs/models/gpt-3-5).

We configure SPAE to encode a $128 \times 128$ image into a token pyramid of 6 layers where each layer has $2^k \times 2^k$ tokens and $k = [0, 1, 2, 3, 4, 4]$. Additionally, we train a video-based SPAE model on the Kinetics-600 dataset [5], and further details can be found in the Appendix. We apply semantic guidance loss to the first five layers, with thresholds of 0.98, 0.95, 0.9, 0.85, and 0.8. The CLIP with a ViT-L/14 [13] vision backbone is used. We use 80 prompt templates from the zero-shot ImageNet

Table 1. **Few-shot classification accuracy** on the mini-ImageNet benchmarks. SPAE_GPT and SPAE_PaLM are trained using different vocabularies and embedding sources, with different prompt templates for in-context learning. They show the broad compatibility of SPAE but are not for a comparison between the LLMs. The best performance with GPT is in italics while the overall best is in bold.

| Method | # Layers : # Tokens | Task Induction Inner Shots Repeats | | ✓ | ✓ | ✓ | ✓ | ✓ | ✓ | Avg |
|---|---|---|---|---|---|---|---|---|---|---|
| | | | 1 0 | 1 0 | 3 0 | 5 0 | 1 1 | 1 3 | 1 5 | |
| *2-Way Classification* | | | | | | | | | | |
| Frozen [35] | - | | 1.7 | 33.7 | 66 | 66 | 63 | 65 | 63.7 | 51.3 |
| LQAE [27] | 1: 256 | GPT 3.5 | 1.5 | 35.2 | 68.2 | 69.8 | 68.5 | 68.7 | 65.9 | 53.97 |
| *SPAE_GPT (ours)* | *2: 5* | *GPT 3.5* | *5.3* | *77.2* | *84.4* | *86.0* | *79.4* | *77.2* | *77.1* | *69.51* |
| *SPAE_PaLM (ours)* | 2: 5 | PaLM 2 | **32.2** | 84.0 | 88.5 | 88.4 | **85.1** | 83.6 | 82.4 | 77.74 |
| *SPAE_PaLM (ours)* | 3: 21 | PaLM 2 | 27.9 | **84.8** | **92.5** | **92.6** | 84.8 | **85.2** | **85.4** | **79.03** |
| *5-Way Classification* | | | | | | | | | | |
| Frozen [35] | - | | 0.9 | 14.5 | 34.7 | 33.8 | 33.8 | 33.3 | 32.8 | 26.26 |
| LQAE [27] | 1: 256 | GPT 3.5 | 1.0 | 15.7 | 35.9 | 36.5 | 31.9 | 36.4 | 45.9 | 29.04 |
| *SPAE_GPT (ours)* | *2: 5* | *GPT 3.5* | *4.3* | *63.0* | *63.4* | *60.6* | *61.9* | *62.1* | *62.1* | *53.91* |
| *SPAE_PaLM (ours)* | 2: 5 | PaLM 2 | **23.6** | 64.2 | 68.0 | 69.9 | 63.4 | 62.0 | 60.2 | 58.76 |
| *SPAE_PaLM (ours)* | 3: 21 | PaLM 2 | 20.2 | **65.1** | **73.7** | **74.3** | **66.4** | **67.0** | **66.3** | **61.86** |

classification setup to precompute the CLIP text embeddings for the vocabulary. In addition, we use the Adam [20] optimizer with loss weights $\alpha = 1, \beta = 0.33, \lambda = 0.1, \eta = 0.1, \phi = 10^{-4}$ and a learning rate of $10^{-4}$ following a linear warmup/cooldown and root square decay schedule. Following the prior work [27], SPAE is trained on the ImageNet ILSVRC2012 [10] dataset. We train with a batch size of 256 for 450k steps. Further details can be found in the Appendix.

### 4.2 Main Evaluation

**Few-shot image classification.** We evaluate the in-context image understanding capability with a frozen LLM on the mini-ImageNet [40] few-shot classification benchmark. A set of tokenized images and class labels are fed to the language model as context for classification of a new image. Following [35, 27], we evaluate 14 settings controlled by four factors regarding the content of each test case: (1) task induction: whether including a preamble to specify the output space; (2) number of ways: the number of categories; (3) number of inner shots: the number of unique examples for each category; (4) number of repeats: the number of times that each unique example is repeated.

We compare SPAE with the state-of-the-art methods Frozen [35] and LQAE [27]. As shown in Tab. 1, SPAE_GPT consistently outperforms LQAE, both using the same GPT 3.5 model and in-context format, while using only 2% of its tokens. Fig. 3 shows the performance trend when using different number of SPAE_PaLM layers across six settings with task induction and 0 repeats. SPAE_PaLM with 3 layers achieves the best performance which balances between sufficient semantics and an image sequence length

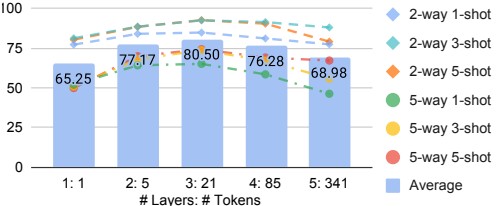

Figure 3. **Few-shot classification accuracy** on mini-ImageNet using different SPAE_PaLM layers.

that is optimal for LLM in-context learning. Overall, SPAE_PaLM yields $+25\%$ and $+32\%$ average accuracy improvement over the state-of-the-art on the 2-way and 5-way benchmarks in Tab. 1.

**Reconstruction quality.** We compare the image and video reconstruction quality using the tokens produced by SPAE and the VQGAN baseline used in state-of-the-art image [7, 25, 6] and video generation [45]. We use FID [16], Inception Score (IS) [33], and LPIPS [48] to compare with the image VQGAN from MaskGIT [7] on the ImageNet validation set, and FVD [36] to compare the 3D-VQGAN from MAGVIT [45] on the Kinetics-600 validation set. The results are presented in Tab. 2. While SPAE may have more lossy reconstruction compared to VQGAN when using a similar number of tokens, this is compensated by going into deeper layers. At the bottom of Tab. 2, we showcase the scalability of our model by training on the ImageNet-21k dataset with 13M images and list the comparable variant from LDM [32] as a reference.

**Token pyramid visualization.** We visualize the tokens produced by SPAE in Fig. 4, where we show the raw pyramid or histogram of tokens with top frequencies for the first four layers, with reconstructed images from layer 5 and 6. We have the following findings.

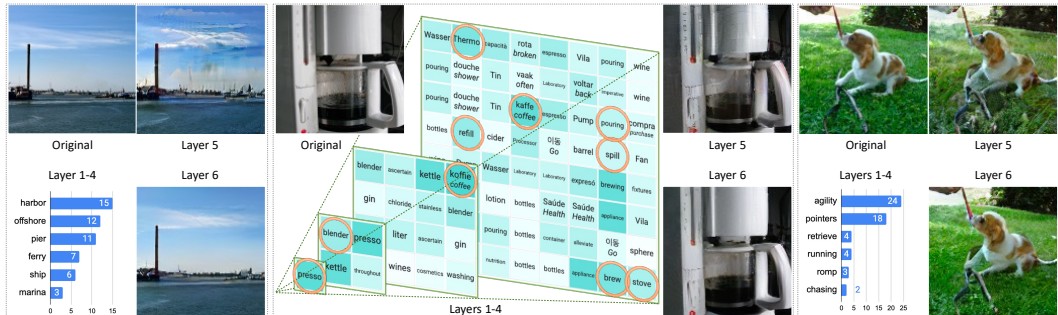

Figure 4. **Examples of pyramid image tokenization and reconstruction** by a 6-layer SPAE. We show the raw pyramid or histogram of most frequent tokens for the first four layers, and reconstructed images from layer 5 and 6. In the pyramid, we use darker cells to show tokens with higher CLIP similarity to the original image. For non-English sub-word tokens, we show automatic translation for reference in italic fonts below the original token. Circled tokens are mentioned in Section 4.2. See full pyramid visualizations in the Appendix.

Table 2. **Comparison of reconstruction quality** between SPAE and VQGAN baselines used in state-of-the-art image [7, 25, 6] and video [45] generation models.

| Resolution | Method | # Layers : # Tokens | Image (ImageNet ILSVRC2012 [10]) | | | Video (Kinetics-600 [5]) |
| | | | FID↓ | IS↑ | LPIPS↓ | FVD↓ |
|---|---|---|---|---|---|---|
| 128×128 | VQGAN | 1: 256 | 5.48 | 119.69 | 0.13 | 6.79 |
| | *SPAE* (ours) | 5: 341 | 9.49 | 109.46 | 0.17 | 52.28 |
| | | 6: 597 | **4.41** | **133.03** | **0.12** | **6.35** |
| 256×256 | VQGAN | 1: 256 | 4.04 | 163.95 | 0.21 | - |
| | *SPAE* (ours) | 6: 597 | **3.60** | **168.50** | **0.19** | - |
| | VQGAN (LDM [32], OpenImages) | 1: 256 | 5.15 | 144.55 | - | - |
| | *SPAE* (ours, ImageNet-21k) | 6: 597 | 3.08 | 173.79 | 0.19 | - |

First, the SPAE tokens are organized in a pyramid structure, with every layer comprising semantically related tokens to the image. The few tokens in the top layers seem to capture the primary theme of the image. For instance, in Fig. 4, the token `presso` (highlighted in orange) represents the espresso machine and other tokens like `blender` refer to related regions. Layer 3 and Layer 4 reveal additional details about localized objects. For example, the token `Thermo` refers to the thermometer in the top-left region, while `stove` appears in the bottom-right area. In addition to nouns, related verbs also show up, including `pouring`, `refill`, `spill`, and `brew`.

Second, it is worth noting that the CLIP model has an English-only vocabulary. However, thanks to the multilingual vocabularies and embeddings from the LLM, SPAE's semantic guidance is able to map to similar concepts in other languages, such as `koffie` in Dutch and `kaffe` in Danish as corresponding terms to the concept of coffee.

Third, similar to RQ tokens [22], SPAE tokens can reconstruct the image with progressively refined details when more layers, and thus tokens, are utilized. Fig. 4 shows Layer 5 begins to produce a reasonable reconstruction while Layer 6 further enhances the level of detail and smoothness.

**Visual question answering.** Tab. 3 provides quantitative results on the visual question answering (VQA) task. We compare with the baseline Frozen [35] method on the Real-Fast-VQA [35] benchmark for few-shot learning. As shown, SPAE consistently outperforms Frozen. Unlike Frozen, SPAE training does not require backpropagation through the LLM.

Table 3. **Few-shot VQA performance** on Real-Fast-VQA.

| Inner Shots | 1 | 3 | 5 |
|---|---|---|---|
| Frozen [35] | 7.8 | 10.1 | 10.5 |
| *SPAE*$_{PaLM}$ (ours) | **14.3** | **15.9** | **15.1** |

### 4.3 Qualitative Studies

This section explores the capability of a frozen PaLM 2, trained solely on language tokens, in performing multimodal tasks using in-context learning. We adopt a two-stage decoding process for image generation. In stage one, we use AR decoding to produce the first 5 SPAE layers with task-specific conditions. Stage two is a task-agnostic NAR decoding process for layer 6 conditioned on the first 5 layers.

**Image to text and VQA.** We examine two tasks involving visual-text reasoning (1) image captioning on COCO [26] captions; and (2) visual question answering (VQA) on COCO-QA [31]. For both

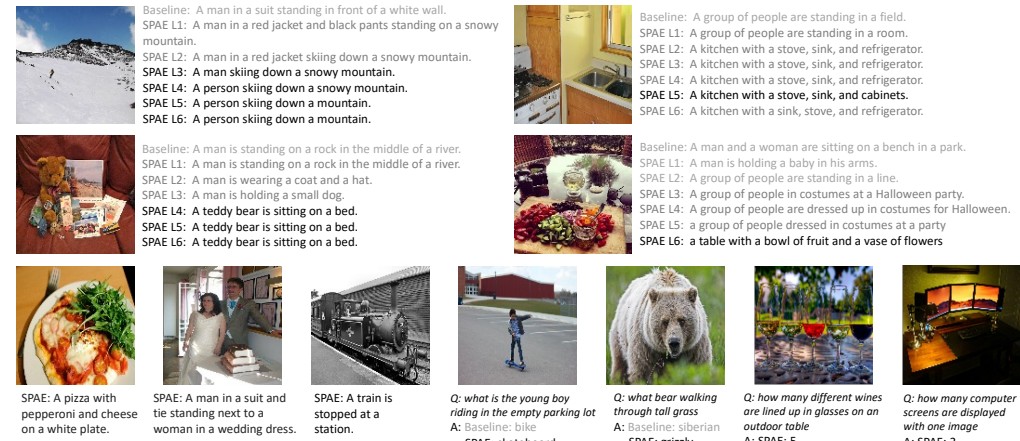

Figure 5. **Qualitative samples of image-to-text generation**: image captioning and VQA. We compare between different layers of SPAE (L1-L6) and a baseline model without semantic guidance or pyramid SAQ.

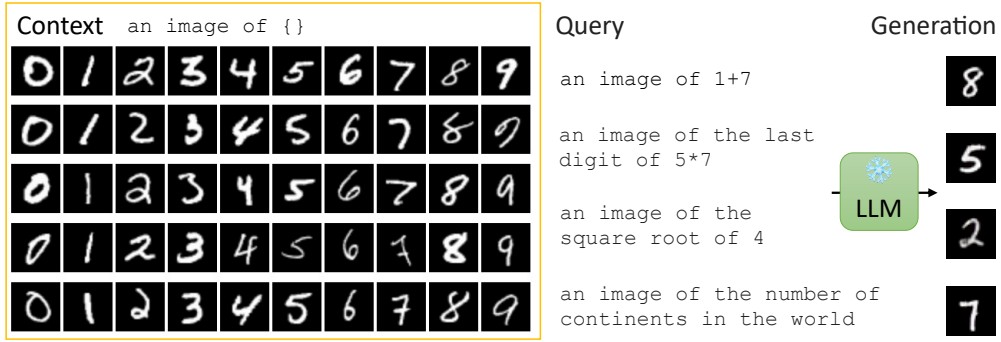

Figure 6. **Examples of text-to-image generation on MNIST** using SPAE with a frozen PaLM 2 model. We use SPAE to tokenize 50 handwritten images as the context and ask PaLM 2, an LLM trained solely on text tokens, to answer complex queries that require generating digit images through SPAE as the output.

tasks, we provide 10 unique training examples as prompts. In the case of VQA, 10 different answers are presented to form a 10-way 1-shot setup.

We compare SPAE to a baseline model trained with the same frozen language codebook but without the proposed semantic guidance or pyramid SAQ. As shown in Fig. 5, when fed with baseline tokens, the LLM randomly hallucinates a caption or guesses an answer simply based on the question. Similar hallucination can happen if we only use the first two layers of SPAE or five words to represent an image, as it provides insufficient context for captioning. Reasonable captions start to appear with 4 layers or 85 words, while complex scenes may still need the full 6 layers of 597 words.

**LLM generating MNIST images.** Fig. 6 shows a few image generation examples on MNIST [11]. The frozen LLM learns about handwritten digit images through 50 context samples tokenized by SPAE trained on MNIST. Each sample consists of a preamble "an image of $k$" and the lexical tokens representing an image of digit $k$. Then we can ask the LLM to answer questions with digit images. Specifically, with a query of "an image of 1+7", we can use progressive AR decoding with the LLM to produce a token sequence that can be decoded into an image of 8 by SPAE. We test with complex questions requiring mathematical reasoning or common sense knowledge, and the LLM is able to respond correctly. In addition, the generated digit images appear different from all context samples. This demonstrates the cross-modal reasoning capability enabled by SPAE and a frozen LLM, with images generated over the text-only interface.

**Conditional image denoising.** To the best of our knowledge, there have been no successful attempts that demonstrate generic image generation capability using a frozen LLM. To this end, we define a simpler denoising setup to explore the capability of LLMs. Fig. 7 demonstrates the conditional image denoising tasks, *e.g.*, image outpainting, deblur, inpainting, location translation, rotation, *etc*. Note that, in order to generate images for each task, we utilize 10 pairs of noisy examples with corruption rates ranging from 50% to 20%, as discussed in Section 3.2. The full context, which is omitted in Fig. 7, can be found in the Appendix.

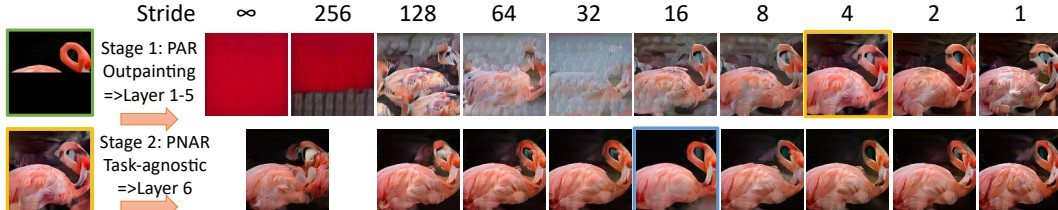

Figure 7. **Examples of conditional image denoising**. We compare different decoding strides for both stages. Yellow and blue boxes indicate the selected results. The LLM is provided with ten pairs of noisy examples like Fig. 2, which are deferred to the Appendix.

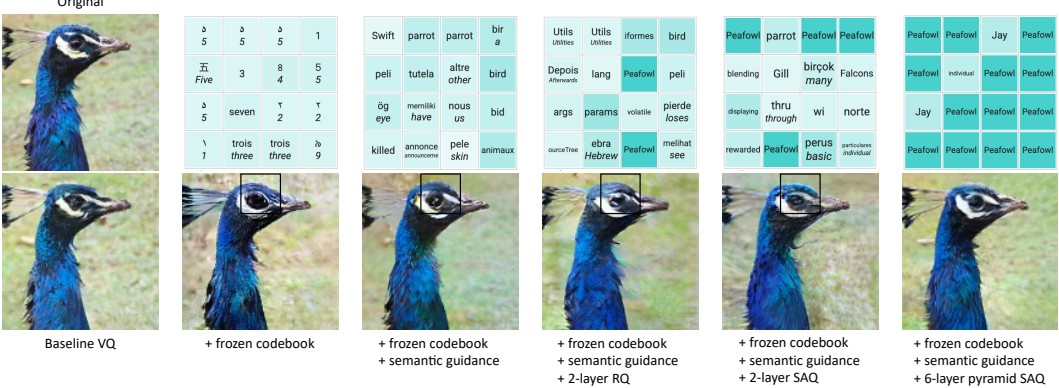

Figure 8. **Ablation examples with reconstructed image and semantic tokens** for models listed in Tab. 4. For non-pyramid tokens, we show a 4×4 crop from the first layer corresponding to the region indicated by the black box. For pyramid tokens, we use the third layer which consists of 4×4 tokens.

The top rows of Fig. 7 compare the generation from different decoding strides with the same set of context examples. Single-step decoding with infinity stride fails to produce a reasonable image, which validates the proposed progressive generation approach.

In Fig. 9, we qualitatively compare SPAE with baseline methods VQGAN and LQAE using the same in-context denoising procedure. As shown, VQGAN fails to produce reasonable images, in part because many words in the LLM output are out of its vocabulary. LQAE only produces vague object contours but cannot recover any visual details. On the contrary, SPAE can generate reasonable images.

**Conditional video denoising and other tasks.** Due to space constraints, we show the examples in the Appendix.

### 4.4 Ablation Studies

The results in Tab. 4 and Fig. 8 verify the effectiveness of the proposed designs in SPAE, as evaluated by reconstruction quality (FID, IS, LPIPS) and semantic relevance (CLIP score,

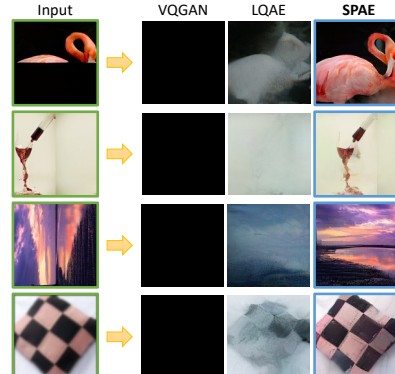

Figure 9. **Comparison on conditional image denoising with different tokenizers**. All models use the same decoding setup with the same ten pairs of prompt images available in the Appendix.

few-shot classification accuracy). We have the following findings. First, simply using a frozen codebook negatively affects the reconstruction results, but with semantic guidance it performs comparably with the original VQGAN while producing meaningful lexical words. Second, RQ hurts reconstruction quality with a frozen codebook. This is different from RQ's standard setup [22] where the codebook is learned. Third, SAQ improves both quality and semantic similarity, where the pyramid enables representation with much fewer tokens. This allows for accommodating more examples within the fixed and constrained in-context length. Finally, per-layer semantic thresholds benefit understanding and the dynamic semantic loss weight helps reconstruction. The perceptual loss leverages a trained network with access to classification labels, but removing it results in a surprising improvement in classification accuracy while greatly hurting the reconstruction.

Table 4. **Ablation studies** on codebook, training objective, quantization method, and token structure. Classification accuracy is evaluated under the mini-ImageNet 5-way 1-shot setup.

| Method | # Layers : # Tokens | FID↓ | IS↑ | LPIPS↓ | CLIP↑ | Classification Accuracy↑ |
|---|---|---|---|---|---|---|
| Baseline VQ | 1: 256 | 5.48 | 119.69 | 0.13 | n/a | 19.6 |
| + frozen codebook | 1: 256 | 7.44 | 101.39 | 0.17 | 0.1464 | 19.5 |
| + semantic loss | 1: 256 | 5.17 | 124.41 | 0.13 | 0.1518 | 46.2 |
| + 2-layer RQ [22] | 1: 256 | 11.94 | 89.01 | 0.22 | 0.1595 | 56.2 |
| | 2: 512 | 6.05 | 113.93 | 0.15 | 0.1547 | - |
| + 2-layer SAQ | 1: 256 | 12.30 | 93.33 | 0.21 | 0.1613 | 56.6 |
| | 2: 512 | 5.08 | 125.27 | 0.14 | 0.1595 | - |
| + 6-layer pyramid SAQ (*SPAE*) | 1: 1 | - | - | - | **0.1879** | 52.0 |
| | 2: 5 | - | - | - | 0.1868 | 64.2 |
| | 3: 21 | - | - | - | 0.1815 | **65.1** |
| | 4: 85 | - | - | - | 0.1711 | 58.5 |
| | 5: 341 | 9.49 | 109.46 | 0.17 | 0.1604 | 46.3 |
| | 6: 597 | **4.41** | **133.03** | **0.12** | 0.1577 | - |
| no per-layer thresholds | 6: 597 | 4.33 | 122.25 | 0.11 | 0.1650 | 59.4 (layer 3) |
| no dynamic semantic weight | 6: 597 | 9.00 | 85.14 | 0.19 | 0.1847 | 65.1 (layer 3) |
| no perceptual loss | 6: 597 | 40.47 | 33.41 | 0.20 | 0.1994 | 69.5 (layer 3) |

# 5 Conclusion

Our work unveils the untapped potential of frozen Large Language Models (LLMs) in tackling multimodal understanding and generation tasks involving images and videos, without requiring explicit training on these modalities. This is achieved by a new method, SPAE, which converts between visual content and lexical tokens of variable length, imbued with rich semantic meaning. Our findings show the great potential of harnessing the vast knowledge and reasoning capabilities of LLMs in the field of computer vision, transcending the limitations of language-only tasks.

**Limitations.** More tokens are required to achieve the same level of reconstruction when using the frozen language codebook, compared to the existing VQGAN models with learned codebooks. The capability of in-context learning is significantly constrained by the acceptable sequence length. Although our results suggest the plausibility of image generation, the resolution, quality, and diversity is far from the recent text-to-image models trained on large image and text data.

**Broader impact.** Our paper showcases the untapped potential of frozen LLMs in multimodal understanding and generation tasks involving images and videos, without requiring explicit training on these modalities. As an initial research proof-of-concept, we focus on in-context learning, which has limitations in learning context and constrained capabilities. Consequently, there is still a substantial gap to the recent specialized models for text-to-image (*e.g.*, Stable Diffusion) or image-to-text that have been specifically trained using billions of text-image pairs.

The potential impact of our research lies in its influence on future studies, specifically in the area of integrating vision modalities into the LLMs. For instance, our work can be extended to explore finetuning or adapter tuning of LLMs on large-scale text-image datasets. Future research in these directions may implicate ethical issues around fairness and transparency. We have found that the generated tokens occasionally include slang terms or words that create inappropriate connotations related to the subject depicted in the image or video. Such concerns must be thoroughly considered and effectively addressed prior to deploying this method in real-world applications.

**Acknowledgments and disclosure of funding.** The authors would like to thank anonymous reviewers and area chairs their insightful comments, and to Siamak Shakeri, Sergey Ioffe, Jay Yagnik, and Boqing Gong for their valuable feedback and constructive discussions. This project is funded in part by Carnegie Mellon University's Mobility21 National University Transportation Center, which is sponsored by the US Department of Transportation.

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
