# SPAE: Semantic Pyramid AutoEncoder
# for Multimodal Generation with Frozen LLMs
# Supplementary Materials

## Appendix Overview

This supplementary document provides additional details to support our main manuscript, organized as follows:

- Appendix A presents more details on the method, including SPAE architecture designs.
- Appendix B provides additional implementation details, including a video SPAE variant.
- Appendix C includes more quantitative evaluation results.
- Appendix D shows more qualitative examples of model generations.

## A Method Details

We present additional details about the SPAE model in this section.

**Token pyramid.** Fig. 10 shows an example of the dilation subsampler defined by Eq. (1). We select evenly distributed positions in each layer to form the token pyramid with monotonically increasing layer sizes.

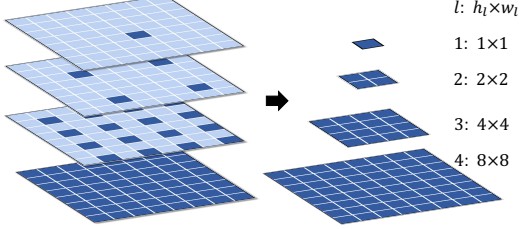

Figure 10. **Dilation subsampler visualization**.

**Streaming average quantization.** Fig. 11 compares our proposed Streaming Average Quantization (SAQ) with Residual Quantization (RQ) [7, 11]. At layer 2, the SAQ remainder embedding $\mathbf{z}_2 = 2\mathbf{z} - \mathbf{e}(k_1)$ is at a more similar scale to $\mathbf{z}$, compared to the RQ remainder $\mathbf{z} - \mathbf{e}(k_1)$. We find that the scale consistency promotes better utilization of the frozen language codebook despite a large number of layers being used. Due to the pyramid structure, quantization in the first few layers may be skipped for those positions not selected by the dilation subsampler. Considering the scale consistency across quantization layers, the use of SAQ is more appropriate in this case.

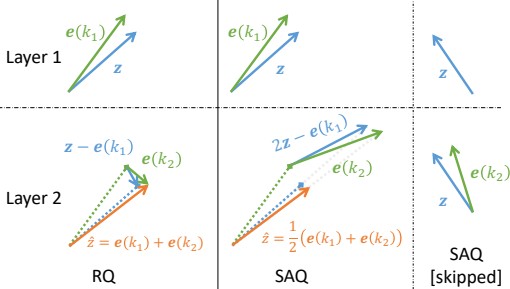

Figure 11. **Comparison between RQ and SAQ.** We show a 2-layer quantization process in a 2-dimensional space as an example. At layer $l$, we use blue for the current remainder embeddings $\mathbf{z}_l$, green for current post-quantization embeddings $\mathbf{e}(k_l)$, and orange for the reconstructed embeddings up to layer $l$ as $\hat{\mathbf{z}}_{\leq l}$.

**In-context denoising.** Take the image-to-image task in Fig. 2 as an example. The provided context are images randomly corrupted in the token space by $\epsilon(\cdot; r)$, where the corruption ratio $r$ follows a cosine schedule [2].

$$(\mathbf{u}^i, \mathbf{v}^i) \sim \left( \epsilon\big(\mathcal{T}(\texttt{mask}(\mathbf{I})); r_i\big), \epsilon\big(\mathcal{T}(\mathbf{I}); r_i\big) \right), \mathbf{I} \in \mathcal{M} \qquad (12)$$

where $\mathcal{T}(\cdot)$ represents the SPAE tokenizer and $\mathcal{M}$ is a small set of raw images. $\texttt{mask}(\cdot)$ zeros out pixels of the real image to create the condition image, such as masking out the bottom half for

37th Conference on Neural Information Processing Systems (NeurIPS 2023).

out-painting. The query $\hat{\mathbf{u}}$ is always sampled from $\mathcal{M}$ without noise $\epsilon$. To ensure the generation is not simply copying the context, we enforce a minimal corruption rate of 20% such that no identical image from the context matches the real target image.

## B   Implementation Details

### B.1   SPAE Training

**Image SPAE.**   An image SPAE encodes a 128×128 image into 16×16 embeddings. Following the VQGAN [5] architecture, we use 128 base filters with channel multipliers [1, 2, 2, 4] and 2 residual blocks at each scale, which results in 59M parameters in total.

**Image SPAE-8.**   In addition to the primary SPAE model with six pyramid layers studied in the main paper, we also train an SPAE-8 model with eight layers to conduct a more in-depth analysis of the coarse-to-fine reconstruction process. The two extra layers each contain 16×16 tokens. The semantic loss is still applied on the first 5 layers as in the primary model.

**MNIST SPAE.**   We train another SPAE on the MNIST [4] dataset with the same architecture setup. We pad the handwritten digit images from 28×28 to 32×32 pixels, which are then encoded into 4×4 embeddings. Each image is represented by 37 tokens organized in four layers, with sizes of 1×1, 2×2, 4×4, and 4×4. We replace the CLIP image embedding with the CLIP text embedding of the label for the semantic loss. The model is trained for 10k steps with a batch size of 256. For in-context generation, AR decoding with a stride of 4 is used to produce all 37 tokens.

**Video SPAE.**   We initialize a video SPAE by VQGAN inflation [10] from a pretrained image SPAE, which encodes 16 frames at 128×128 resolution into 4×16×16 embeddings. A video SPAE consists of 176M parameters. The pyramid layers contain 1×1×1, 1×2×2, 1×4×4, 2×8×8, 4×16×16, and 4×16×16 tokens. The video embedding is obtained as the average CLIP embedding for all frames. The model is trained on the Kinetics-600 [1] dataset which contains 384k videos. We train with a batch size of 512 for 130k steps, which takes 5.8k TPUv4-hours.

### B.2   LLM Prompting

To generate prompts, we utilize SPAE to quantize an image, or another non-linguistic modality, into a pyramid of lexical tokens. Subsequently, we flatten the tokens by concatenating them layer-by-layer, following a raster scan, and resulting in a 1-D string. This string, representing the image, is referred to as the ***SPAE string*** in the following prompts.

We use task-specific prompt templates to facilitate answer generation with LLMs. The LLM output is always parsed by removing leading and trailing whitespace or newline characters.

**Image classification with GPT 3.5.**   We use the same prompt template as LQAE [8] to interact with GPT 3.5. For a 2-way 1-shot classification between class *lion* and *vase*, the prompt is

```
For each of the following input output pairs, output is one of ['lion', 'vase']
###
Input: <SPAE string from a lion image>
Output: lion
###
Input: <SPAE string from a vase image>
Output: vase
###
Input: <SPAE string from the query image>
Output:
```

We use greedy decoding to get a maximum of 7 tokens from GPT 3.5.

**Image classification with PaLM 2.**   We use the original miniImageNet [9] format with PaLM 2. The prompt looks like

```
Answer with "lion" or "vase".

<SPAE string from a lion image>
This is a lion

<SPAE string from a vase image>
This is a vase

<SPAE string from the query image>
What is this?  # Only used in 5-way 3/5-shot setups
This is a
```

We use greedy decoding to get a maximum of 4 tokens from PaLM 2.

**Image captioning.**    We use greedy decoding to get a maximum of 20 tokens before the first newline character with the following prompt:

```
Generate a caption sentence based on words describing an image.

Q: <SPAE string from image 1>
A: <Caption for image 1>

Q: <SPAE string from image 2>
A: <Caption for image 2>

Q: <SPAE string from the query image>
A:
```

**Visual question answering.**    We use greedy decoding to get a maximum of 4 tokens before the first newline character with the prompt template as

```
Answer with a single word.

C: <SPAE string from image 1>
Q: <Question for image 1>
A: <Answer for image 1>

C: <SPAE string from image 2>
Q: <Question for image 2>
A: <Answer for image 2>

C: <SPAE string from the query image>
Q: <Question for the query image>
A:
```

**Image/video generation with PAR decoding.**    For image or video generation tasks, the condition can be a text string or an SPAE string of a condition image. Suppose we use PAR decoding with a stride of 4 tokens. At the 4th step, the prompt looks like

```
Learn a new language and predict the 4 tokens following the examples.

C:<condition for image 1>
Q:<SPAE string (token 1-12) for image 1>
A:<SPAE string (token 13-16) for image 1>

C:<condition for image 2>
Q:<SPAE string (token 1-12) for image 2>
A:<SPAE string (token 13-16) for image 2>
```

```
C:<condition for the query>
Q:<SPAE string (token 1-12) for the generated image from previous steps>
A:
```

We use PaLM 2 to generate 8 predicted sequences for the next 4 tokens, starting with a temperature $T_0 = 0$. We use the sentence piece [6] tokenizer to tokenize the output string. If all predictions are shorter than 4 tokens, we retry the LLM prediction with a higher temperature. At the $i$-th retry, the temperature is given by

$$T_i = \psi \sum_{j=1}^{i} 2^j \tag{13}$$

where $\psi = 0.01$ is used.

**Image/video generation with PNAR decoding.**   We use PNAR decoding to generate SPAE layer 6 conditioned on layer 1-5. With a stride of 16, the prompt at the 3rd step looks like

```
Predict the outputs following the examples.

Q:<SPAE string from layer 1-5 for image 1>
A:<SPAE string from layer 6 (token 33-48) for image 1>

Q:<SPAE string from layer 1-5 for image 2>
A:<SPAE string from layer 6 (token 33-48) for image 2>

Q:<SPAE string from layer 1-5 for the generated image from AR decoding>
A:
```

We use PaLM 2 to generate 8 predicted sequences for the next 16 tokens. If the sentence piece parsing fails, we retry with the same temperature schedule as in PAR decoding.

### B.3   Corruption Functions

**Pixel-space transformation.**   We use pixel-space transformation in the conditional image interpolation tasks with the following setups:

- Brightness: $[\pm 0.8, \pm 0.6, \pm 0.4, \pm 0.2]$.
- Contrast: $[\pm 0.8, \pm 0.6, \pm 0.4, \pm 0.2]$.
- Saturation: $[\pm 0.4, \pm 0.3, \pm 0.2, \pm 0.1]$.
- Color (RGB): $[(0.6, 1.4, 1), (0.7, 1.3, 1), (0.8, 1.2, 1), (0.9, 1.1, 1),$
  $(1.1, 0.9, 1), (1.2, 0.8, 1), (1.3, 0.7, 1), (1.4, 0.6, 1)]$

Overflow pixels are clipped to $[0, 255]$.

**Token-space permutation noise.**   Random permutation is used in the in-context denoising setup for conditional image denoising tasks. Specifically, we replace a fraction of tokens each with a random token sampled from the entire 65k vocabulary to satisfy a given corruption rate. The corruption rates for the 10 examples are $[0.5, 0.47, 0.44, 0.41, 0.38, 0.35, 0.32, 0.29, 0.26, 0.23]$. The permutation noise presents a context distribution with expectation at the real image, but does not contain the ground truth tokens to prevent information leakage.

## C   Additional Quantitative Results

**Few-shot image classification with different SPAE layers.**   Tab. 5 present the few-shot mini-ImageNet classification performance with each SPAE$_{\text{PaLM}}$ layer. These detailed quantitative numbers accompany the findings from Fig. 3. As shown, Layer 3 achieves the best overall performance as well as in most of the setups, which balances between the level of details and the burden of the LLM.

Table 5. **Few-shot classification accuracy** on the mini-ImageNet benchmarks. - means value unavailable due to an infeasible sequence length.

| Method | # Layers : # Tokens | Task Induction Inner Shots Repeats |  | ✓ | ✓ | ✓ | ✓ | ✓ | ✓ | Avg |
|---|---|---|---|---|---|---|---|---|---|---|
|  |  |  | 1 | 1 | 3 | 5 | 1 | 1 | 1 |  |
|  |  | PaLM 2 | 0 | 0 | 0 | 0 | 1 | 3 | 5 |  |
| *2-Way Classification* |  |  |  |  |  |  |  |  |  |  |
| $SPAE_{PaLM}$ | 1: 1 | PaLM 2 | **34.8** | 77.2 | 81.2 | 80.3 | 74.0 | 73.2 | 71.5 | 70.31 |
| $SPAE_{PaLM}$ | 2: 5 | PaLM 2 | 32.2 | 84.0 | 88.5 | 88.4 | **85.1** | 83.6 | 82.4 | 77.74 |
| $SPAE_{PaLM}$ | 3: 21 | PaLM 2 | 27.9 | **84.8** | **92.5** | **92.6** | 84.8 | **85.2** | **85.4** | **79.03** |
| $SPAE_{PaLM}$ | 4: 85 | PaLM 2 | 22.8 | 81.1 | 91.4 | 90.4 | 82.6 | 84.3 | 84.7 | 76.76 |
| $SPAE_{PaLM}$ | 5: 341 | PaLM 2 | 21.2 | 77.4 | 88.0 | 79.1 | 84.8 | 74.0 | 76.1 | 71.51 |
| $SPAE_{PaLM}$ | 6: 597 | PaLM 2 | 21.8 | 73.8 | 70.8 | 62.4 | 64.8 | 62.1 | 58.6 | 59.19 |
| $SPAE_{PaLM}^{disjoint}$ | 2: 5 | PaLM 2 | 24.8 | 79.8 | 84.5 | 83.7 | 80.8 | 78.5 | 78.4 | 72.93 |
| $SPAE_{PaLM}^{disjoint}$ | 3: 21 | PaLM 2 | 21.4 | 81.4 | 89.2 | 87.9 | 82.6 | 81.7 | 80.6 | 74.98 |
| *5-Way Classification* |  |  |  |  |  |  |  |  |  |  |
| $SPAE_{PaLM}$ | 1: 1 | PaLM 2 | **26.8** | 52.0 | 50.9 | 49.9 | 51.9 | 48.4 | 47.9 | 46.83 |
| $SPAE_{PaLM}$ | 2: 5 | PaLM 2 | 23.6 | 64.2 | 68.0 | 69.9 | 63.4 | 62.0 | 60.2 | 58.76 |
| $SPAE_{PaLM}$ | 3: 21 | PaLM 2 | 20.2 | **65.1** | **73.7** | **74.3** | **66.4** | **67.0** | 66.3 | **61.86** |
| $SPAE_{PaLM}$ | 4: 85 | PaLM 2 | 16.1 | 58.5 | 67.2 | 69.1 | 64.0 | 66.4 | **67.4** | 58.39 |
| $SPAE_{PaLM}$ | 5: 341 | PaLM 2 | 12.1 | 46.3 | 55.9 | 67.2 | 43.3 | 46.3 | - | - |
| $SPAE_{PaLM}$ | 6: 597 | PaLM 2 | 12.1 | 35.7 | - | - | - | - | - | - |

Table 6. **Reconstruction quality and semantic relevance** of SPAE-8 tokens.

| Model | # Layers : # Tokens | FID↓ | IS↑ | LPIPS↓ | CLIP↑ | Relative CLIP↑ |
|---|---|---|---|---|---|---|
|  | 1: 1 | - | - | - | **0.2051** | **0.8018** |
|  | 2: 5 | - | - | - | 0.2046 | 0.7994 |
|  | 3: 21 | - | - | - | 0.2012 | 0.7834 |
| SPAE-8 | 4: 85 | - | - | - | 0.1896 | 0.7289 |
|  | 5: 341 | 43.42 | 49.78 | 0.32 | 0.1709 | 0.6412 |
|  | 6: 597 | 8.93 | 116.12 | 0.18 | 0.1667 | 0.6213 |
|  | 7: 853 | 4.78 | 135.01 | 0.13 | 0.1647 | 0.6119 |
|  | 8: 1109 | **3.89** | **140.55** | **0.11** | 0.1634 | 0.6058 |

**Few-shot image classification with SPAE$^{disjoint}$.** Following the previous work of LQAE [8], we train our SPAE on the ImageNet training split [3] and present the comparative results in the main paper. There is a possibility of overlap between the training split of ImageNet and the mini-ImageNet dataset used in the few-shot classification task [9]. Since few studies have investigated this before, we present the results of training SPAE on the ImageNet training split after excluding the 20 classes used in the few-shot mini-ImageNet classification task. This creates a even more challenging setting as the visual classes have never been seen during the training of the tokenizer or the LLMs.

As demonstrated in Tab. 5, we present the results of training our tokenizer on the *disjointed* data, referred to as SPAE$^{disjoint}$. As expected, we observe a slight decrease in performance, since both SPAE and LLMs need to generalize to the test classes that are outside the training data distribution. Despite the fact that the baseline is trained on unlabeled images sampled from the mini-ImageNet test classes, SPAE$_{PaLM}^{disjoint}$ still demonstrates a significant improvement over the state-of-the-art baseline on the 2-way benchmarks.

**Token quality with more SPAE layers.** Tab. 6 shows the per-layer reconstruction quality and semantic relevance of tokens from the SPAE-8 model in comparison to the default model. With more token layers, the model gains larger capacity for both semantic and appearance, where the appearance gets pushed into deeper layers. At layer 1 to 6, SPAE-8 yields consistently higher CLIP scores than SPAE. At the last three layers, SPAE-8 also has better reconstruction quality than the last two layers

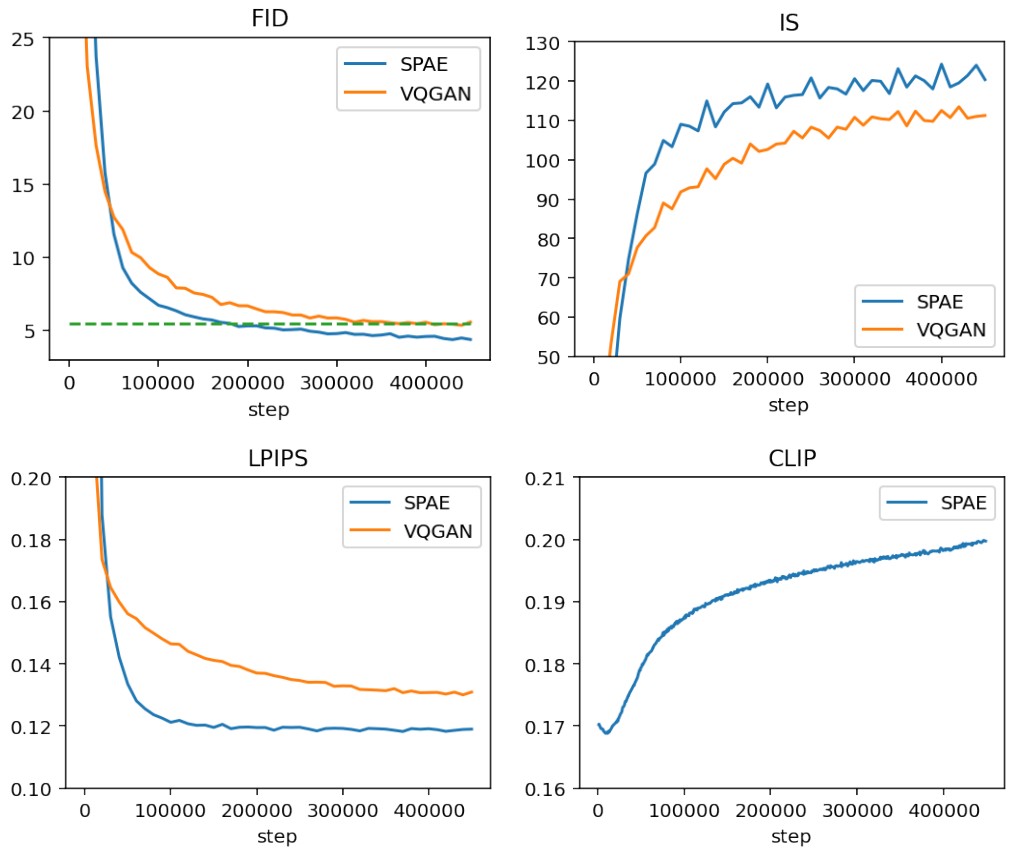

Figure 12. **Training curves** of SPAE in comparison to VQGAN. Metrics are presented regarding reconstruction quality (FID, IS, LPIPS) and semantic relevance (CLIP).

of SPAE. These results suggest the potential of better reconstruction quality and semantic relevance from using more token layers.

**Training efficiency.** All models used in the ablation study in Tab. 3, including VQGAN [5] and RQ-VAE [7] variants, are trained using the same setup for fair comparisons. Fig. 12 compares the training curves of FID, IS, LPIPS, and CLIP score of SPAE and VQGAN. As shown, within 40% of the training steps, SPAE shows better FID than the final VQGAN checkpoint. The CLIP score keeps improving as the training proceeds, while the LPIPS saturates quite early.

## D   Additional Qualitative Examples

**Token pyramid visualization.** Fig. 13 shows tokenization and reconstruction samples by a 6-layer SPAE from ImageNet validation set. Key concepts are captured in the first few layers, whereas the later layers focus on the visual appearance. In the coffee machine example, many keywords are present to describe various aspects from the stove to the thermometer. In the parrot case, a single unified concept is repeatedly highlighted.

**Coarse-to-fine reconstruction.** Fig. 14 shows reconstruction samples by SPAE-8 from ImageNet validation set. We compare the reconstructed images from layer 5 to layer 8 to demonstrate the coarse-to-fine progress.

**Conditional image interpolation.** To the best of our knowledge, there have been no successful attempts that demonstrate generic image generation capability using a frozen LLM. To this end, we define a very simple setup to explore the interpolation capability of LLM, where the conditions are

integers from 1 to 9. The target images are created with different pixel-space transformations detailed in . As shown in Fig. 15, images 1-4 and 6-9 are fed as context to produce image 5, where the model interpolates the variable property. Fig. 16 shows generated samples at $256\times256$ resolution under the same setup.

**Conditional image denoising.** We use PAR decoding to produce the first 5 token layers with task-specific conditions, followed by task-agnostic PNAR decoding to fill in layer 6. Fig. 17 visualizes the input pairs for the image-to-image generation examples in Figs. 7 and 9, with more examples in Fig. 18. Under the in-context denoising setup, the LLM generates novel images based on the provided context, where multiple different generations can be obtained.

**Multimodal outputs.** Fig. 19 shows a task requiring a single LLM to output both image and text, where it first inpaints the center region of an image using in-context denoising and then creates multiple captions for the output image.

**Image-to-video denoising.** Fig. 20 shows an image-to-video example with the frame prediction task using progressive in-context denoising. The input is one frame tokenized by the image SPAE, while the output is a 16-frame clip tokenized by the video SPAE. We follow the same two-stage procedure as image-to-image generation, with more steps in each stage to account for the longer sequence. Due to the sequence length limit, only four samples can be fit into the context, which limits LLM's performance for this task.

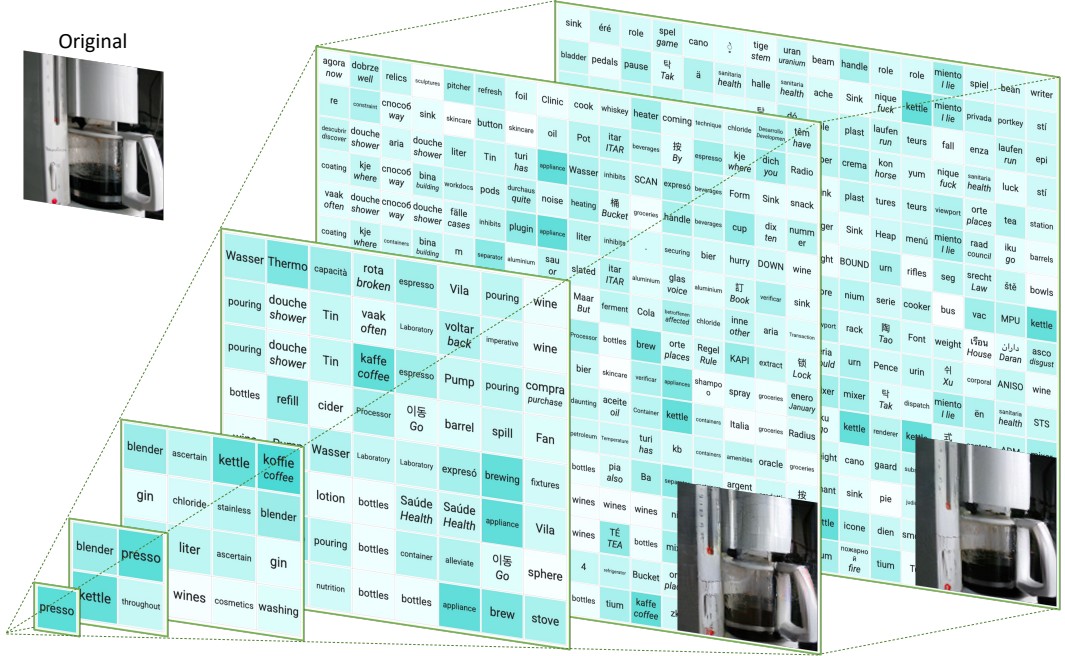

(a) Many keywords are present to describe various aspects from the stove to the thermometer.

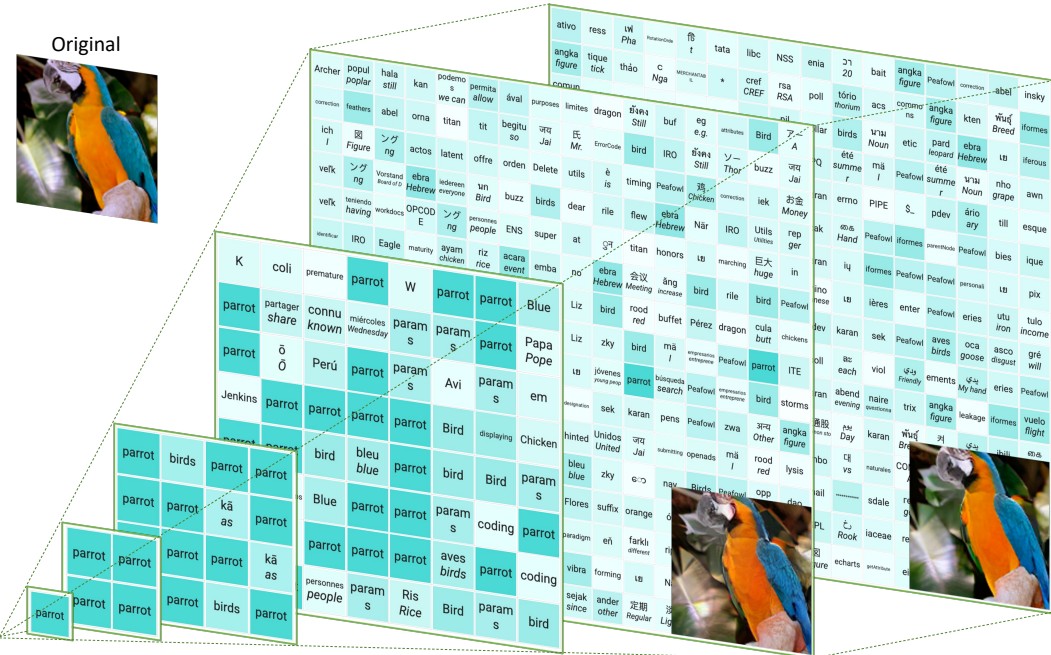

(b) A single unified concept is repeatedly highlighted.

**Figure 13. Examples of multi-layer image tokenization and reconstruction** by a 6-layer SPAE. For visualization purposes only, we use darker cells to show tokens with higher CLIP scores regarding the original image. For non-English sub-word tokens, we show automatic translation for reference in italic fonts below the original token. We show tokens in all six layers, along with reconstructed images from the last two layers.

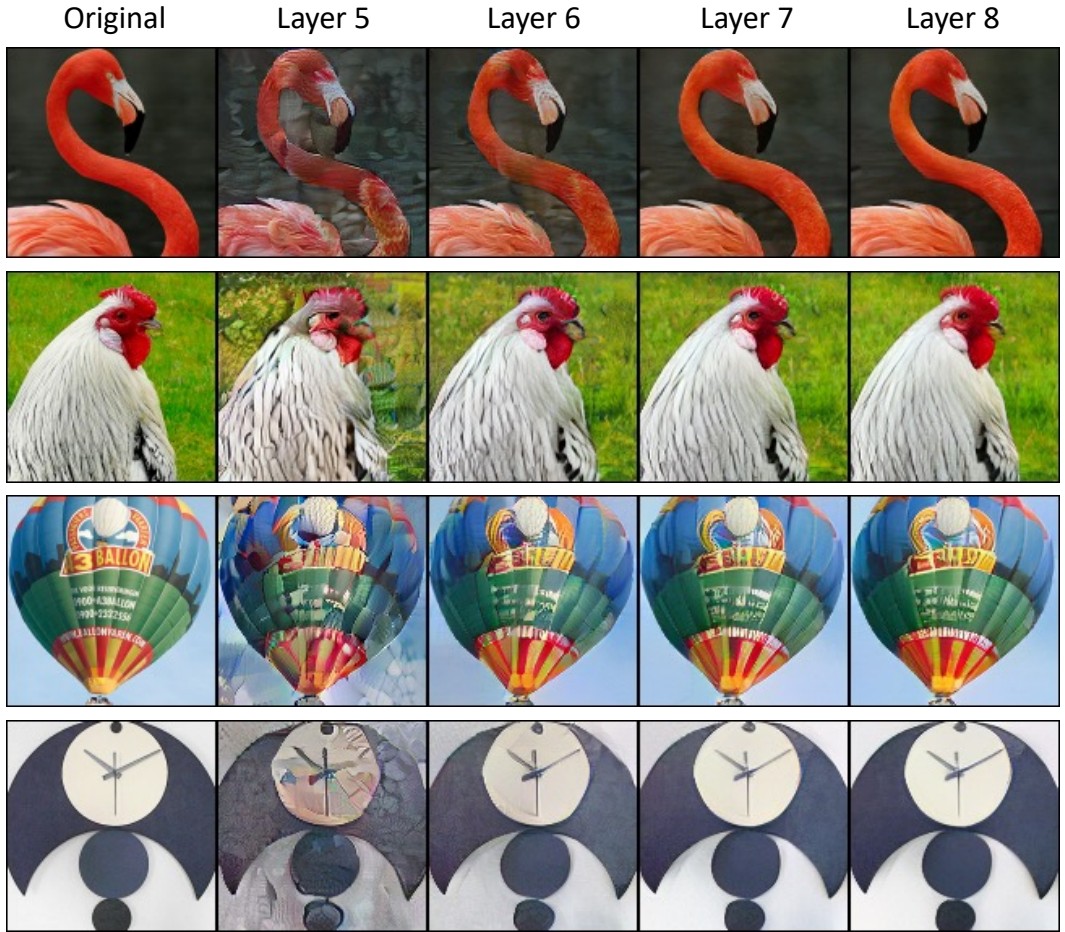

Figure 14. **Examples of coarse-to-fine image reconstruction** by SPAE-8. The top 5 layers reconstruct a noisy image. The appearance details gradually get refined as more token layers are aggregated by the streaming average quantization process.

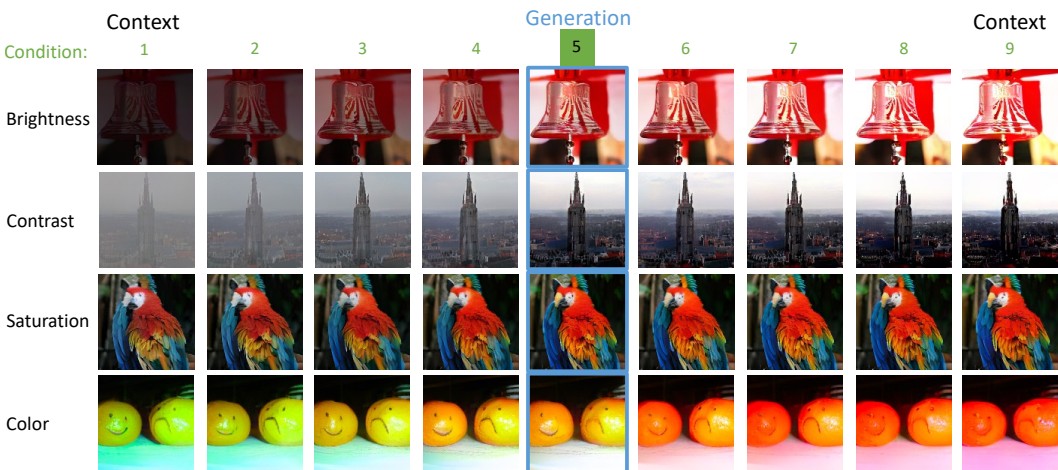

Figure 15. **Examples of conditional image interpolation** of different image transformations.

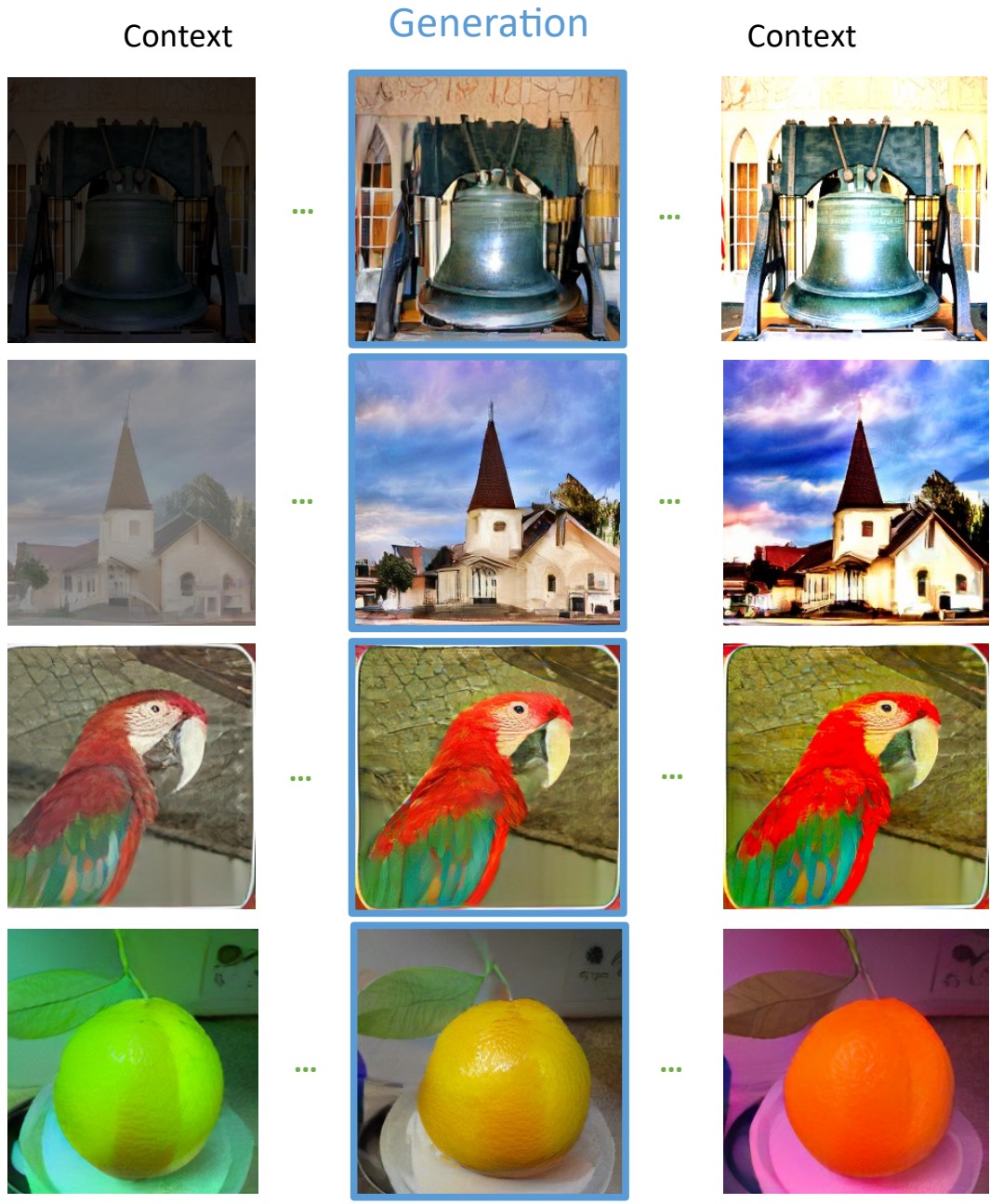

Figure 16. **Examples of conditional image interpolation** at 256x256 resolution. The LLM is provided with eight condition images for the interpolation following the setup in Fig. 15.

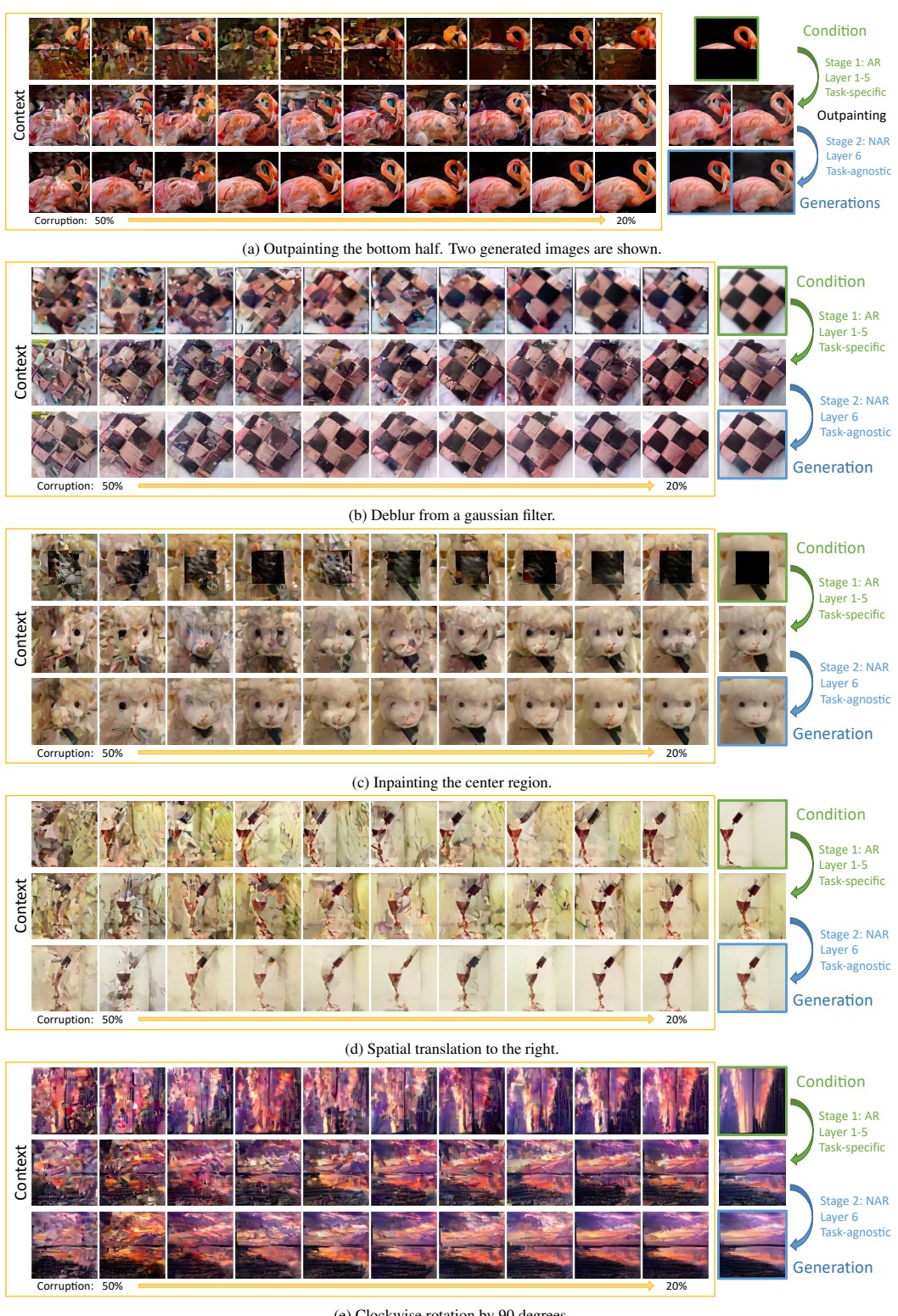

(a) Outpainting the bottom half. Two generated images are shown.

(b) Deblur from a gaussian filter.

(c) Inpainting the center region.

(d) Spatial translation to the right.

(e) Clockwise rotation by 90 degrees.

Figure 17. **Examples of conditional image denoising**. All input samples for the in-context learning are presented for the examples in Figs. 7 and 9. The LLM generates novel images based on the provided context. Multiple different generations can be obtained from the same set of context samples.

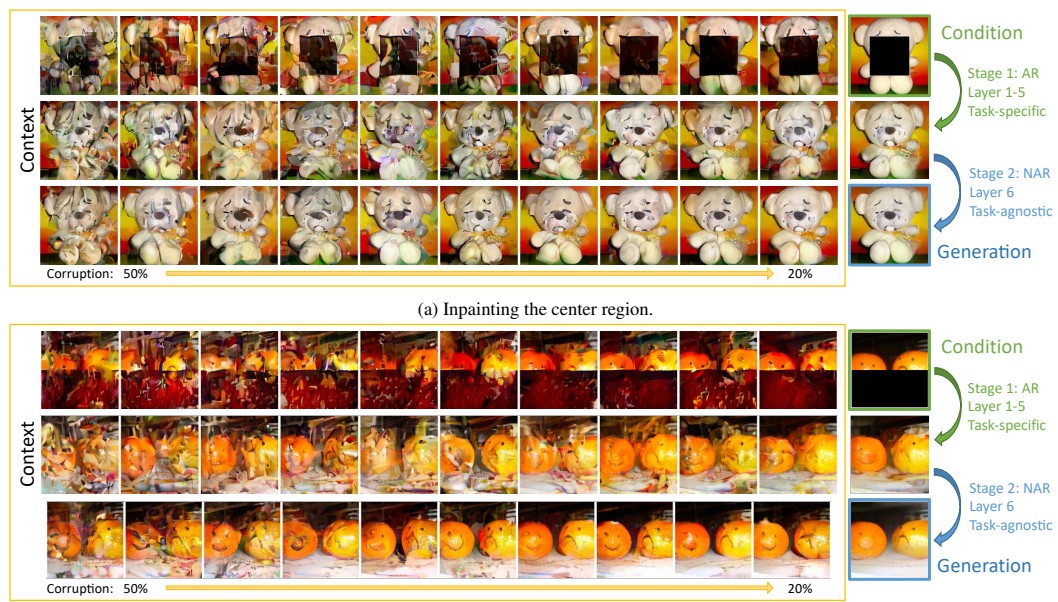

(a) Inpainting the center region.

(b) Outpainting the bottom half.

Figure 18. **More examples of conditional image denoising**. The LLM generates novel images based on the provided context image pairs.

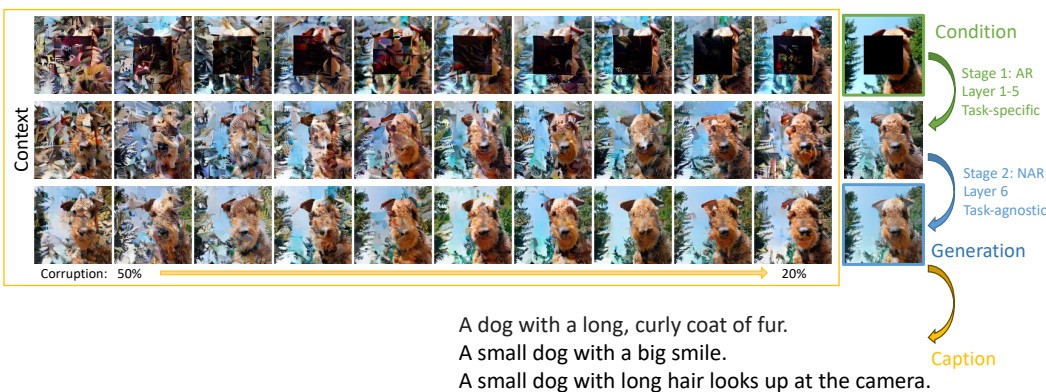

A dog with a long, curly coat of fur.
A small dog with a big smile.
A small dog with long hair looks up at the camera.

Figure 19. **Examples of multimodal outputs** from the LLM. The LLM generates a novel image with multiple captions based on the provided context.

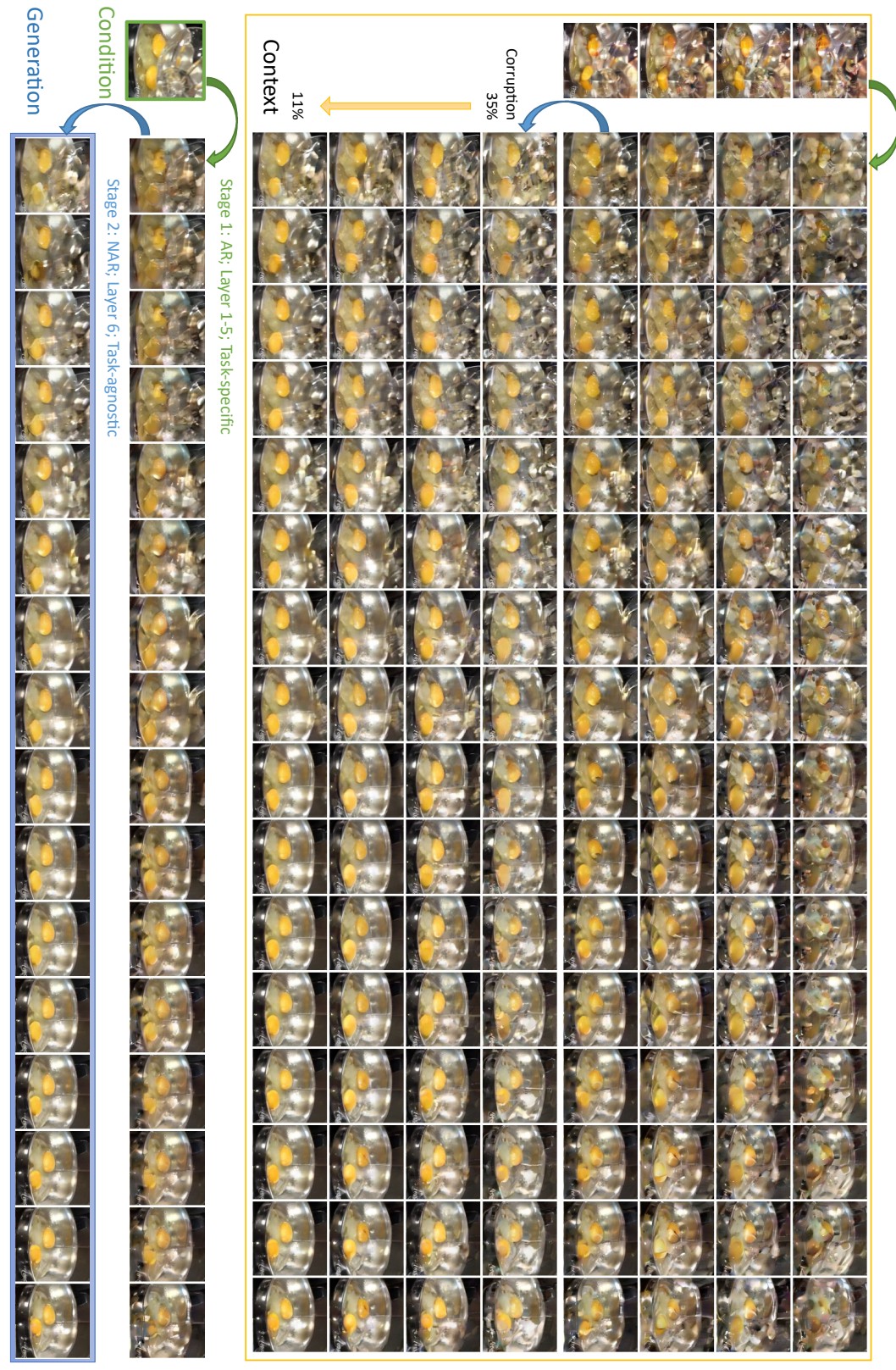

Figure 20. **Examples of image-to-video denoising**: frame prediction. We follow the same two-stage generation procedure as in image-to-image tasks. Due to the sequence length limit, only four samples can be fit into the context. The generated video clip appear visually different from the context samples, especially around the reflections of the bowl.