# OpenReview forum: "SPAE: Semantic Pyramid AutoEncoder for Multimodal Generation with Frozen LLMs"
_NeurIPS.cc/2023/Conference — NeurIPS 2023 spotlight_

### Official Review · Reviewer_1ht3 · 2023-07-05

**Soundness:** 3 good
**Presentation:** 2 fair
**Contribution:** 4 excellent
**Rating:** 7
**Confidence:** 4

**Summary:**

This paper introduces the Semantic Pyramid AutoEncoder (SPAE) for enabling frozen Large Language Models (LLMs) to perform understanding and generation tasks involving non-linguistic modalities such as images or videos. SPAE converts raw pixels to lexical tokens extracted from the LLM's vocabulary, allowing the LLM to perform multimodal tasks. The proposed method is evaluated using in-context learning experiments with frozen PaLM 2 and GPT-3.5, demonstrating success in image understanding and generation tasks. The main contributions include the first successful method using a frozen language model for image content generation, a new SPAE tokenizer producing interpretable representations, a new progressive prompting method for long cross-modal sequences, and evaluation on visual understanding and generation tasks.

**Strengths:**

1. Compared to LQAE, the method presented in this paper demonstrates superior performance in image reconstruction.

2. By interacting with large language models such as GPT-3.5 and PaLM-2, this paper effectively utilizes the in-context learning capabilities of these models to generate corresponding images, offering an intriguing research direction.

**Weaknesses:**

1. This paper's experiments are limited to 128x128 resolution, significantly lower than the mainstream 256x256 clarity.

2. To achieve comparable performance with VQGAN, a larger latent space (# tokens) is necessary.

3. Is it possible to attain the same performance as VQGAN + frozen codebook + semantic guidance and an expanded latent space (# tokens)? The need for a pyramid structure comes into play here. Whether the pyramid structure is necessary requires experimental support.

4. The VAE proposed in this paper employs perceptual loss and utilizes a VGG network with supervised pre-training on ImageNet during the training process. In contrast, other methods like LQAE do not use perceptual loss. A strategy similar to ViT-VQGAN should be adopted here, using a version without perceptual loss for training on the comprehension task.

5. There is a lack of model variants in the ablation experiments on the understanding tasks.

6. In lines 203-204, it is mentioned that "We train with a batch size of 256 for 450k steps, which takes 1.4k TPUv3-hours." This training cost is larger than that of both VQGAN and RQ-VAE. It would be interesting to investigate the impact of the training cost on the performance of the VAE proposed in this paper.

**Questions:**

Figure 8 in the appendix is intriguing, but the experiments are only conducted on the MNIST dataset. It would be compelling to see the experimental results on datasets like MS-COCO.

**Limitations:**

addressed

---

> ### Author Rebuttal · Authors · 2023-08-09
>
> We appreciate the detailed comments from reviewer 1ht3 and the acknowledgment of our contributions as “the first successful method using a frozen language model for image content generation,” where our method “demonstrates superior performance in image reconstruction” and offers “an intriguing research direction.”
> We respond to each question below.
>
> > W1: 256x256 resolution.
>
> We train an SPAE model to tokenize 256x256 images with a downsampling factor of 16, which yields the same number of tokens as 128x128. As shown in Table R4.1, SPAE outperforms VQGAN in reconstruction FID. More importantly, unlike the VQGAN tokens, SPAE tokens carry semantic information, as measured by the CLIP score, and have been demonstrated to benefit understanding tasks (see Table 1 in the paper).
>
> In Fig. 17 in [the rebuttal pdf](https://openreview.net/attachment?id=sIoDsBBe1G&name=pdf), we visualize the in-context conditional image generation at 256x256.
>
> **Table R4.1 Comparison of reconstruction quality and semantic relevance at different resolutions.**
>
> |Resolution|Method|FID↓|IS↑|LPIPS↓|CLIP↑|
> |-|-|--|-|-|-|
> |128x128|VQGAN|5.48|119.69|0.13|n/a|
> ||SPAE|4.41|133.03|0.12|0.1577|
> |256x256|VQGAN|4.04|163.95|0.21|n/a|
> ||SPAE|3.60|168.50|0.19|0.1637|
>
>
> > W2: SPAE needs larger latent space than VQGAN.
>
> As recognized by reviewer ekNF and mRhs, SPAE tokens capture “both semantic and fine-grained visual details.” Compared to VQGAN, which only captures visual details, more bandwidth is expected to store the additional semantic information while achieving on-par or better reconstruction quality.
>
>
> > W3: is it possible to attain the same performance as VQGAN + frozen codebook + semantic guidance? The necessity of pyramid structure?
>
> Comparing the third model (VQGAN + frozen codebook + semantic guidance) and SPAE (6-layer pyramid SAQ), Tab. 4 and Tab. R4.2 show that SPAE performs better in both understanding (accuracy 65.1 vs. 46.2) and reconstruction (FID 4.41 vs. 5.17). In particular, SPAE uses significantly fewer tokens to represent an image (using 21 vs. 256 tokens) for the understanding task. The high token efficiency results from the pyramid structure because a conventional layer without pyramid structures needs a minimum of 256 tokens to represent the image. The pyramid design facilitates the representation of images using much fewer tokens, a vital factor for in-context learning as it enables the accommodation of more examples within the provided context. We will clarify this point in the revised version.
>
> In Fig. 19 in [the rebuttal pdf](https://openreview.net/attachment?id=sIoDsBBe1G&name=pdf), SPAE shows better reconstruction quality and more relevant tokens than the third model.
>
> **Table R4.2 Ablation results on miniImageNet classification with PaLM 2, supplementary to Tab. 4.**
>
> |Method|#Layers:#Tokens|CLIP↑|5-way 1-shot Accuracy↑|
> |-|-|-|-|
> |VQGAN|1:256|n/a|19.6|
> |+frozen codebook|1:256|0.1464|19.5|
> |++semantic guidance (the third model)|1:256|0.1518|46.2|
> |+++2-layer RQ|1:256|0.1595|56.2|
> |+++2-layer SAQ|1:256|0.1613|56.6|
> |+++6-layer pyramid SAQ|3:21|0.1815|**65.1**|
>
>
> > W4: removing perceptual loss for understanding tasks.
>
> As suggested, we train an SPAE variant without perceptual loss. As shown in Tab. R4.3, removing the perceptual loss greatly hurts the reconstruction quality as expected but results in even better classification accuracy. We hypothesize that this could be attributed to the fact that, compared to the standard model, this variant focuses much less on appearance and allocates more learning capacity for semantics, which leads to better understanding performance. Nevertheless, the results in Table R4.3 show that SPAE consistently outperforms the baseline LQAE when no perceptual loss is used.
>
> **Table R4.3 Ablation study on the perceptual loss with ImageNet reconstruction and mini-ImageNet classification (3 token layers, PaLM 2).**
>
> ||FID↓|IS↑|2-way Avg Accuracy↑|5-way Avg Accuracy↑|
> |-|-|-|-|-|
> |LQAE|||53.97|29.04|
> |SPAE|**4.41**|**133.03**|79.03|61.86|
> |SPAE w/o perceptual loss|39.83|33.23|**79.81**|**64.89**|
>
>
> > W5: performance of ablated model variants on understanding tasks.
>
> As suggested, we evaluate all model variants in the ablation experiments (Tab. 4) on the miniImageNet 5-way 1-shot image classification benchmark. These results are shown in Tab. R4.2. We will add these results in the revised version.
>
>
> > W6: performance with regard to training cost.
>
> We would like to mention that all models used in the ablation study in Tab. 4, including VQGAN and RQ-VAE variants, are trained using the same setup for fair comparisons.
> In Fig. 20 in [the rebuttal pdf](https://openreview.net/attachment?id=sIoDsBBe1G&name=pdf), we show the training curves of FID, IS, LPIPS, and CLIP score of SPAE and VQGAN.
> As shown, within 40% of the training steps, SPAE shows better FID than the final VQGAN checkpoint.
> The CLIP score keeps improving as the training proceeds, while the LPIPS saturates quite early.
>
>
>  > Q1: Figure 8 in the appendix is intriguing; what about the experimental results on datasets like MS-COCO?
>
> Thanks for the encouraging comments regarding Fig 8. As we acknowledged in L312-313, “the capability of in-context learning is significantly constrained by the acceptable sequence length” of the LLM. Dealing with the complex scenes on MS-COCO falls beyond the paper’s scope of “in-context learning” because it needs the finetuning of LLMs on a larger dataset. We hope our paper could facilitate future studies to explore using SPAE to jointly finetune LLMs on larger text-image training datasets to generate images with enhanced fidelity.

---

> > ### Comment · Reviewer_1ht3 · 2023-08-19
> > **Re: Rebuttal by Authors**
> >
> > Overall, the reviewer is satisfied with the author's response, particularly the experimental results presented in Table R4.1 and Table R4.2. The author is requested to include these elements in the final version of the manuscript. The reviewer has increased the rating.

---

### Official Review · Reviewer_2BCt · 2023-07-05

**Soundness:** 3 good
**Presentation:** 3 good
**Contribution:** 3 good
**Rating:** 6
**Confidence:** 4

**Summary:**

This paper proposes Semantic Pyramid AutoEncoder (SPAE) which enables frozen LLMs to perform both understanding and generation tasks involving non-linguistic modalities such as images or videos through in-context learning. The authors first introduce SPAE, a pyramid tokenizer that produces interpretable representations of semantic concepts and fine-grained details. In addition, the authors also propose a new progressive prompting method that facilitates the in-context generation of long cross-modal sequences. The proposed method outperforms SoTA in few-shot image classification accuracy by a large margin.

**Strengths:**

- The proposed approach does not require further tuning of LLM. It is thus very portable.
- The paper is well-written and the approach, motivations, and experiments are well-derived and clear.
- The paper presents an interesting algorithm to leverage LLMs to address non-linguistic tasks, such as how to combine pretrained models and how well it would perform, even without tuning LLM and its codebook.
- Impressive few-shot results on image classification.


**Weaknesses:**

Although this paper has shown how a frozen LLM can generate image content or realize visual understanding tasks. The tasks the authors choose to evaluate in this paper are still limited (only few-shot image classification on ImageNet).
Can the authors provide more quantitative experiments to validate the SPAE’s generalization to various tasks (e.g., caption, VQA…)? Besides, there are only qualitative examples of image-to-image generation, which only prove that SPAE “can” generate image content, not that it can generate well.

**Questions:**

please refer to the weakness part.

**Limitations:**

The authors have described the limitations of this work.

---

> ### Author Rebuttal · Authors · 2023-08-09
>
> We thank reviewer 2BCt for the thoughtful comments. Please see our response below.
>
> > W: quantitative results on VQA.
>
> Following the reviewer’s suggestion, we provide quantitative results on the VQA task in Tab. R3.1.
> We compare with the baseline Frozen [35] method on the Real-Fast-VQA benchmark for few-shot learning. As shown, SPAE consistently outperforms Frozen. It is worth mentioning that, unlike Frozen, SPAE training does not require backpropagation through the language model.
>
> **Table R3.1 Few-shot VQA performance on Real-Fast-VQA.**
>
> | Inner Shots   | 1    | 3    | 5    |
> |---------------|------|------|------|
> | Frozen [35]        | 7.8  | 10.1 | 10.5 |
> | SPAE | 14.3 | 15.9 | 15.1 |
>
>
> > W: generation capability.
>
> Our contribution is a proof-of-concept that, with SPAE, an LLM trained on text tokens “can” indeed generate image content. This marks the first successful attempt to enable a frozen LLM to generate image content through in-context learning. To the best of our knowledge, there have been no standard evaluation setups for such in-context image generation. In Fig. 18 in [the rebuttal pdf](https://openreview.net/attachment?id=sIoDsBBe1G&name=pdf), we qualitatively compare in-context denoising generation results with our baseline methods VQGAN and LQAE. The results validate that SPAE is the only method capable of generating reasonable image content through in-context learning. We will clarify this point in the revised version.

---

> > ### Comment · Reviewer_2BCt · 2023-08-18
> >
> > Thank you for the authors' responses. They have addressed my comments. I therefore keep my initial positive rating.

---

> > > ### Author Response · Authors · 2023-08-18
> > > **Thank you.**
> > >
> > >
> > > Thank you for the comments. As we have addressed your concerns, we appreciate it if you consider raising your score.

---

### Official Review · Reviewer_mRhs · 2023-07-07

**Soundness:** 3 good
**Presentation:** 3 good
**Contribution:** 3 good
**Rating:** 5
**Confidence:** 4

**Summary:**

This paper presents a novel multi-modal generation method named SPAE which leverages semantic autoencoder and in-context denoising for semantic reconstruction. The SPAE converts raw pixels into interpretable lexical tokens from the LLM's vocabulary, capturing both semantic and fine-grained visual details. The paper conducts in-context learning experiments using frozen PaLM-2 and GPT 3.5 on a diverse set of image understanding and generation tasks for showing the effectiveness.

**Strengths:**

● Semantic Pyramid AutoEncoder (SPAE) is introduced for enabling frozen LLM to perform both understanding and generation tasks involving non-linguistic modalities, such as images or videos.
● Different from previous work, SPAE produces interpretable representations of semantic concepts and fine-grained details in the form of multilingual linguistic tokens by leveraging frozen LLM.
● The proposed method outperforms the best-published few-shot image classification accuracy by an absolute 25% under the same in-context setting.

**Weaknesses:**

1. The 5-shot evaluation on ImageNet classification is somewhat unfair. It is not clear about the performance with GPT-3.5 under this setting. The paper only reports the results of PaLM-2.
2. The hyperparameters in the paper are quite tricky (e.g. threshold for each layer, the weight of the loss).  Besides, the effect of dynamic weight used in the final objective is not ablated.
3. Since SPAE is only trained on ImageNet, it is not unknown whether this method scales well on large-scale datasets (e.g. Laion-400M, Laion-2B, COYO-700M, etc).

**Questions:**

See weakness

**Limitations:**

The paper demonstrates the generation under the low-resolution setting. The higher resolution (e.g. 1024x1024, 2048x2048) remains unexplored. Can you provide some comparisons with diffusion, such as high resolution and complex scenes?

---

> ### Author Rebuttal · Authors · 2023-08-09
>
> We appreciate the constructive comments from reviewer mRhs and the assessment that our method “produces interpretable representations” and “outperforms the best-published few-shot image classification.”
> We respond to each question below.
>
> > W1: Missing 5-way miniImageNet with GPT 3.5?
>
> Following the reviewer’s suggestion, we add the 5-way classification on the mini-ImageNet benchmark in Tab. R2.1, which compares SPAE$\_\text{GPT}$ and LQAE [27] with “the same GPT 3.5 model and in-context format” (L217-218). The results are consistent with the observation in Table 1 in the paper that SPAE$\_\text{GPT}$ significantly outperforms LQAE.
>
> **Table R2.1.  5-way classification accuracy on the mini-ImageNet benchmark with GPT 3.5. The results are supplementary to Tab. 2 in the paper.**
>
> |                       |         |          |          |          |          |          |          |           |
> |-----------------------|---------|----------|----------|----------|----------|----------|----------|-----------|
> | Task Induction        |         | yes      | yes      | yes      | yes      | yes      | yes      | Avg       |
> | Inner Shots           | 1       | 1        | 3        | 5        | 1        | 1        | 1        |           |
> | Repeats               | 0       | 0        | 0        | 0        | 1        | 3        | 5        |           |
> | LQAE [27]                | 1       | 15.7     | 35.9     | 36.5     | 31.9     | 36.4     | 45.9     | 29.04     |
> | **SPAE$_\text{GPT}$** | **4.3** | **63.0** | **63.4** | **60.6** | **61.9** | **62.1** | **62.1** | **53.91** |
>
>
> > W2: hyperparameters (e.g., threshold for each layer, the weight of the loss) and ablation on the effect of dynamic weight.
>
> It’s worth highlighting that most weight hyperparameters in the training loss Eq. (8) closely follow the hyperparameters in the previous works [44, 12]. Following the reviewer’s suggestion, we discuss the two new hyperparameters presented in this paper: the layer-wise threshold and the dynamic weight used in Eq. (9).
>
> Regarding the layer-wise threshold, we train SPAE using a uniform threshold for every layer. As shown in Tab. R2.2, this variant attains comparable reconstruction quality but yields worse understanding performance. The result indicates that layer-wise thresholding might not be absolutely necessary, but it appears beneficial for understanding tasks.
>
> To analyze the impact of the dynamic weight, we train an SPAE variant using a constant weight of 1 instead. As shown in Tab. R2.2, removing the dynamic weight results in an imbalance between semantics and appearance, which leads to much worse reconstruction quality. The results substantiate the contribution of dynamic weight for the generation task.
>
> We will clarify these points in the revised version.
>
> **Table R2.2 Ablation experiments on training hyperparameters. We evaluate reconstruction quality on ImageNet, along with miniImageNet 5-way 1-shot classification accuracy with PaLM 2.**
>
> |                               | Reconstruction |        |        | Understanding |
> |-------------------------------|----------------|--------|--------|---------------|
> | Method                        | FID↓           | IS↑    | LPIPS↓ |  5-way 1-shot Accuracy↑ |
> | SPAE  (Baseline)                        | 4.41           | 133.03 | 0.12   | 65.1      |
> | &nbsp; &nbsp; uniform threshold  | 4.33           | 122.25 | 0.11      | *59.4*      |
> | &nbsp; &nbsp; constant weight              | *9.00*           | *85.14*  | *0.19*    | 65.1      |
>
>
> > W3: scalability to larger datasets.
>
> To test its scalability, we train SPAE at 256x256 resolution on a much larger dataset ImageNet-21k which consists of 13M images and is 10x larger than ImageNet. We selected ImageNet-21k because we could not acquire copyright permission to train on the datasets recommended by the reviewer.
>
> The results in Tab. R2.3 show the model performance on the standard ImageNet validation set. These results provide empirical evidence of the SPAE model’s scalability across a large dataset of 13 million images from ImageNet21k.
> In Tab. R2.3, the performance in the Latent Diffusion Model (LDM) paper [32] is also listed for reference. Note that direct comparison to LDM might not be appropriate due to using a distinct dataset (OpenImages).
>
> In addition, we train SPAE on Kinetics-600, which consists of 384k videos with around 96M frames. The results are in Tab. 3 in the paper and Fig. 16 in the supplementary material.
>
> **Table R2.3 Comparison of reconstruction quality and semantic relevance on ImageNet-val.**
>
> | Method   | Resolution | Dataset      | FID↓ | IS↑    | CLIP↑  |
> |----------|------------|--------------|------|--------|--------|
> | SPAE     | 128x128    | ImageNet     | 4.41 | 133.03   | 0.1577 |
> | SPAE     | 256x256    | ImageNet-21k | 3.08 | 173.79   | 0.1637 |
> |          |            |              |      |        |        |
> | LDM [32] | 256x256    | OpenImages   | 5.15 | 144.55   | n/a    |
>
>
> > L1: higher resolution such as 1024x1024.
>
> As we acknowledged in L312-313, “the capability of in-context learning is significantly constrained by the acceptable sequence length” of the LLM. In practice, the largest resolution we tried is 256x256, as shown in Fig. 17 of [the rebuttal pdf](https://openreview.net/attachment?id=sIoDsBBe1G&name=pdf).
>
> It is worth highlighting that our method is just “the first successful attempt to enable a frozen LLM to generate image content” (L10-11). This point has also been acknowledged by Reviewer 1ht3. Scaling to higher resolutions or comparing with diffusion models trained on extensive image-text datasets falls beyond the paper’s scope, necessitating further research. For example, future studies could explore using SPAE to jointly finetune LLMs on larger text-image training datasets to generate high-resolution images and complex scenes. We will clarify this point in the revised version.

---

> > ### Comment · Reviewer_mRhs · 2023-08-17
> >
> > I appreciate the authors' response. Their rebuttal addressed part of my concerns. This work is quite interesting, but whether it can generalize to scaling and high resolution still needs to be validated. Therefore, I still choose to maintain my previous score.

---

> > > ### Author Response · Authors · 2023-08-17
> > > **Request for Elaboration: Thanks for the Feedback – Kindly Seek Clarification on Reviewer's Comment**
> > >
> > > We are thankful for the reviewer’s feedback and assessment that our “work is quite interesting”.
> > >
> > > We would appreciate if the reviewer could elaborate on the comment  “whether it can generalize to scaling and high resolution still needs to be validated”.
> > >
> > > We have presented the results of **256x256 images using SPAE trained on ImageNet21K (~13 million images)** (see Table R2.3 in the rebuttal)
> > >
> > >
> > > **[Scaling]**  We have demonstrated that training SPAE on 13 million images can be completed within a matter of 1-2 days . The results may already provide substantial validation of scalability given that training on larger datasets such as Laion-400M would merely entail increasing the training time by a factor of 30 or deploying additional GPU/TPU devices.
> > >
> > >
> > > **[High resolution]** We would like to provide more clarifications.  In the text-to-image models (e.g., Parti, MUSE, Latent Diffusion), the native image generation resolution has typically been 256x256. To obtain high-resolution, such as 1024x1024, it would require separate super-resolution models. In Fig. 17 of [the rebuttal pdf](https://openreview.net/attachment?id=sIoDsBBe1G&name=pdf), we have presented the results of 256x256 images which can be used as inputs to the super-resolution model. Therefore, our results may validate that our method can scale to high resolution images as incorporating super-resolution apparently falls beyond the scope of our paper
> > >
> > >
> > > We kindly request the reviewer to carefully assess whether we have addressed the concern raised regarding scaling and high resolution. Please let us know for additional questions.
> > >
> > > Thank you

---

### Official Review · Reviewer_ekNF · 2023-07-07

**Soundness:** 3 good
**Presentation:** 3 good
**Contribution:** 3 good
**Rating:** 6
**Confidence:** 3

**Summary:**

The paper introduces SPAE, a method that aligns visual representation with a fixed LLM representation. SPAE effectively captures semantics and visual fine-grained textures for a range of cross-modality tasks. The paper showcases good and solid results in few-shot learning and reconstruction/generation tasks.






**Strengths:**

- The paper is well written and well motivated.
- The proposed idea is interesting, cross-modality with codebook is not a novel idea, but using the fixed pre-trained LLM as dictionary is an interesting idea and seems working well.
- The results are solid and convincing.

**Weaknesses:**

- The quantitive comparisons are not fair comparisons; as the language model sued in SPAE is much bigger and trained on much more data than the baselines. One interesting baseline is to replace the heavy LLM to smaller LMs and discuss the impact.
- Lack of comparisons to other works on generation tasks. It would be good to include comparison to other generative models with same prompt.

**Questions:**

- Is it possible to perform zeroshot tasks with the visual encoder and LLM, it would be a good example to compare SPAE with other contrastive based models.
- How critical is the LLM used in this paper, can we switch LLM with other language models and how will that impact the performance?

---

> ### Author Rebuttal · Authors · 2023-08-09
>
> We thank reviewer ekNF for the detailed feedback. Please see our response below.
>
>
> > W1: fairness of the quantitative comparisons due to the LM size.
>
> We kindly note that this seems to be a misunderstanding. In Table 1, both SPAE$\_\text{GPT}$ and the baseline LQAE [27] are based on **the same GPT 3.5 model and in-context format** (as mentioned at L217-218 in the manuscript). Our approach outperforms the baseline LQAE by an improvement of over 15\% when utilizing the identical language model.
>
> Nevertheless, to eliminate the LLM dependency during SPAE training, we also train a SPAE$\_\text{RoBERTa}$ model, which uses precisely the same embedding as in LQAE. Under the same in-context evaluation protocol with GPT 3.5 for miniImageNet 5-way 1-shot classification, SPAE$\_\text{RoBERTa}$ achieves 36.5\% accuracy, which is significantly better than LQAE with 15.7\%. This result further validates the effectiveness of the SPAE over the baseline LQAE.
>
>
> > W2: comparisons on generation tasks.
>
> “Our work marks the first successful attempt to enable a frozen LLM to generate image content” (L10-11) through in-context learning, a point acknowledged by other reviewers. Therefore, it is challenging to directly compare to other methods under the same setting. To overcome this challenge, we compare SPAE to the baselines (VQGAN [12], LQAE) using the same prompt images and progressive in-context denoising (proposed in Sec. 3.2).
>
> The comparison is shown in Fig. 18 in [the rebuttal pdf](https://openreview.net/attachment?id=sIoDsBBe1G&name=pdf). As shown, VQGAN fails to produce reasonable images, mainly because many words in the LLM output are out of its vocabulary. LQAE only produces vague object contours but cannot recover any visual details. On the contrary, SPAE can generate better images.
>
>
> > Q1: Is it possible to perform zero-shot tasks with the visual encoder and LLM?
>
> The LLMs used in our paper are trained on text-only data. This differs from the zero-shot cross-modal methods, such as contrastive-based models, that are trained on large-scale image-text data. Consequently, the employed LLMs do not understand images, making it non-trivial to perform zero-shot cross-modal tasks. Our paper shows this can be compensated by providing a few image examples via in-context learning.
>
>
> > Q2: LLM flexibility.
>
> “Our method has been tested with PaLM 2 and GPT-3.5, suggesting compatibility with arbitrary LLMs.” (L46-47) Per L186-194, each language model uses different tokenizers with different vocabularies, and we extract word embeddings for SPAE from different layers of the model. These setup variations further suggest the compatibility of our approach.

---

> > ### Comment · Reviewer_ekNF · 2023-08-16
> > **Thanks for the rebuttal**
> >
> > The work is very interesting, and the rebuttal addressed most of my concerns.

---

> > > ### Author Response · Authors · 2023-08-16
> > > **Thank you for the comments**
> > >
> > > Thank you for the comments. We appreciate that you would consider increasing the rating as all concerns are addressed.

---

### Author Rebuttal · Authors · 2023-08-09

Dear reviewers, ACs, and SACs,

Thank you for your time in handling this submission. We thank all reviewers for their thoughtful comments. We appreciate the reviewers’ acknowledgment that our idea/algorithm is perceived as “interesting” (ekNF, 2BCt). Our method SPAE is “the first successful method using a frozen language model for image content generation” (1ht3), produces “interpretable representations” (mRhs, 2BCt, 1ht3), captures “both semantic and fine-grained visual details” (mRhs, eKnf), and is “very portable” (2BCt). Regarding the result, the reviewer's comments highlight that our method “outperforms SoTA in few-shot image classification accuracy by a large margin” (2BCt), “outperforms the best-published few-shot image classification accuracy by an absolute 25%” (mRhs), “demonstrates superior performance in image reconstruction” (1ht3), and is “solid and convincing” (ekNF).

We provide detailed responses below each review, with additional evidence and explanations. The PDF file for the figures is attached to this general response.

---

### Author Response · Authors · 2023-08-15
**A Kind Reminder on Reviewer-Author Discussions**

Dear Reviewers,

Thanks again for your effort and valuable feedback during the review process. We have made our responses and hope they address your concerns. Please let us know if you have further questions after reading our rebuttal.

We hope to address all the potential issues during the discussion period.

Best,

Authors

---

### Decision · Program_Chairs · 2023-09-21

**Decision:**

Accept (spotlight)

**Comment:**

The reviewers find the proposed approach interesting, especially the idea of using a pre-trained LLM’s vocabulary as the codebook making the model interpretable. The paper presents the first approach to use Frozen LLM for image generation. The reviewers are intrigued by this research direction. They also appreciate the computationally efficiency of the proposed approach since it does not require fine-tuning of LLMs. The reviewers are impressed by the experimental results. They found the paper to be well written and well motivated.

The reviewers had raised some concerns, but the rebuttal successfully addressed most of them and all reviewers recommend acceptance. The authors are encouraged to improve the final paper version by following reviewer recommendations.